



# Dominant Patterns of Summer Ozone Pollution in Eastern China and Associated Atmospheric Circulations

Zhicong Yin [12], Bufan Cao[1], Huijun Wang[12]

[1]Key Laboratory of Meteorological Disaster, Ministry of Education / Joint International Research Laboratory of Climate and
Environment Change (ILCEC) / Collaborative Innovation Center on Forecast and Evaluation of Meteorological Disasters
(CIC-FEMD), Nanjing University of Information Science & Technology, Nanjing 210044, China
[2]Nansen-Zhu International Research Centre, Institute of Atmospheric Physics, Chinese Academy of Sciences, Beijing, China

*Correspondence to*: Zhicong Yin(yinzhc@163.com)

**Abstract.** Surface ozone, a man-made air pollutant, has been severe during summers in the eastern parts of China, damaging
human's health and flora and fauna. During 2015–2018, ground-level ozone pollution increased year by year and intensified
from south to north. In North China and Huanghuai region, the $O_3$ concentrations were highest. Two dominant patterns of
summer ozone pollution were determined, i.e., a south-north covariant pattern and a south-north differential pattern. The
anomalous atmospheric circulations composited for the first pattern manifested as a zonally enhanced East Asia deep trough
and as a west Pacific subtropical high whose western ridge point shifted northward. The local hot, dry air and intense solar
radiation enhanced the photochemical reactions to elevate the $O_3$ pollution levels in North China and Huanghuai region. For
the second pattern, the broad positive geopotential height anomalies at high latitudes significantly weakened cold air activity,
and those extending to North China resulted in locally high temperature near the surface. In a different manner, the west Pacific
subtropical high transported sufficient water vapor to the Yangtze River Delta and resulted in locally adverse environment for
the formation of surface ozone. Furthermore, the implications for the interannual differences in summer $O_3$ pollution have also
proven to be meaningful.

## 1.   Introduction

Ozone occurs both in the stratosphere and at ground level. Stratospheric ozone forms a protective layer that shields us from the
sun's harmful ultraviolet radiation. However, surface ozone is a man-made air pollutant and has harmful effects on people and
on the environment, such as damaging human lungs (Day et al., 2017)and destroying agricultural crops and forest vegetation
(Yue et al., 2017). Worldwide, severe ozone events are more frequent and stronger in China than those that have taken place in
Japan, South Korea, Europe, and the United States (Lu et al. 2018). Due to their close relationship with anthropogenic
emissions (Li et al., 2018), the high $O_3$ concentrations in China are mainly observed in urban regions, such as in North China
(NC), the Yangtze River Delta (YRD) and the Pearl River Delta (PRD) where rapid development has occurred in recent
decades (Wang et al., 2017). An increase in surface ozone levels was found in China in 2016 and 2017 relative to 2013 and
2014 (Lu et al., 2018). The O3 pollution levels in Beijing-Tianjin-Hebei (part of NC) were the most severe in China (Wang et





al., 2006; Shi et al., 2015) and this situation has been getting worse. The $O_3$ concentrations in North China underwent a significant increase in the period of 2005–2015, with an average rate of $1.13 \pm 0.01$ ppb $yr^{-1}$ (Ma et al., 2016). Even on the highest mountain over NC, Mount Tai, summer (June-July-August, JJA) O3 increased significantly by 2.1ppbv $yr^{-1}$ (Sun et al., 2016). The $O_3$ levels generally presented increasing trends from 2012 to 2015 in the YRD (Tong et al., 2017), e.g., the $O_3$

concentrations in Shanghai (a mega-city) increased by 67% from 2006 to 2015 (Gao et al., 2017). In the PRD region, $O_3$ increased by 0.86 ppbv $yr^{-1}$ from 2006 to 2011 (Li et al., 2014). Severe ozone pollution is projected to increase in the future over eastern China (Wang et al., 2013).

Although deep stratospheric intrusions may elevate surface ozone levels (Lin et al., 2015), the main source of surface ozone is the photochemical reactions between the oxides of nitrogen ($NO_x$) and volatile organic compounds (VOC), i.e., $NO_x$ + VOC =

$O_3$. The concentrations of $NO_x$ and VOC are fundamental drivers impacting ozone production, and are sensitive to the regime of ozone formation, i.e., $NO_x$-limited or VOC-limited (Jin and Holloway 2015). The changes in fine particulate matter are also a pervasive factor for the variation in ozone concentration. Li et al (2018) found that rapid decreases in fine particulate matter levels significantly stimulated ozone production in NC. Furthermore, the meteorological conditions also influenced the ozone levels via modulation of the photochemical episodes (Yin et al., 2019). Violent solar radiation accelerated chemical $O_3$

production (Tong et al., 2017). A severe heat wave in the YRD contributed to high $O_3$ concentrations in 2013 (Pu et al., 2017). Winds had an impact on the $O_3$ and its precursors at downwind locations (Doherty et al., 2013). Local meteorological influences are always related to specific large-scale atmospheric circulations. The changes in the East Asian summer monsoon led to 2–5% interannual variations in surface $O_3$ concentrations over central eastern China (Yang et al., 2014). Continental anticyclones created sunny and calm weather, which are favourable conditions for $O_3$ production in NC (Ding et al., 2013; Yin

et al., 2019). Due to their large-scale descending motion, tropical cyclones are often related to the evaluation of surface $O_3$ levels in the PRD (Ding et al., 2004). Further studies showed that a strong west Pacific subtropical high (WPSH) was unfavourable for the formation of $O_3$ in South China, due to the association with more clouds, more rainfall, decreased ultraviolet radiation and lower air temperatures (Zhao and Wang, 2017).

Wang et al (2017) reviewed the meteorological influences on ozone events, but the referenced findings were published mainly

before 2010, when measurements in China were still scarce. Since 2015, $O_3$ measurements in eastern China were steadily and widely implemented, but the $O_3$-weather studies mainly focused on several synoptic processes. Nevertheless, the dominant patterns of summer ozone in east of China are still unclear, which is the topic of this study. The findings of this study basically help to understand the features of surface ozone pollution in eastern China, their relationships with large-scale atmospheric circulations and the implications for the climate variability.



**2.    Data sets**

Public hourly O₃ concentration data since May 2014 are available on the website (http://beijingair.sinaapp.com/). Although the summer data from 2014 can be downloaded, the observations of atmospheric compositions were beginning to be uniformly distributed and continuously achieved in eastern China since 2015. After processing missing values, 868 sites in eastern China (110°E–127°E, 22°N–45°N) from 2015 to 2018 were employed here to reveal the features of surface ozone pollution and its

associated mechanisms. The maximum daily average 8 h concentration of ozone (MDA8) is the maximum of the running 8 h mean O₃ concentration during an entire 24 hour day. According to the Technical Regulation on Ambient Air Quality Index of China (the Ministry of Environmental Protection of China, 2012), MDA8 is generally used to represent the O₃ conditions. The 2.5 °×2.5 °ERA-Interim data used here include the geopotential height (Z), zonal and meridional wind, relative humidity, vertical velocity, air temperature at different pressure levels, surface air temperature (SAT) and wind, downward solar

radiation at the surface, low and medium cloud cover and precipitation (Dee et al. 2011). Because the maximum photochemical activity often occurred at afternoon (Wang et al., 2010), the daytime data (i.e., 08 a.m.–08 p.m. Beijing time) were calculated by the sub-daily reanalysis to composite the daytime atmospheric circulations and daytime meteorological conditions. In addition, the monthly ERA-Interim reanalysis was also employed here to discuss the impacts of atmospheric circulations on interannual differences or the climate variability of O₃ pollutions in eastern China.

**3.    Variations and dominant patterns**

During 2015–2018, summer surface ozone pollution was severe in China, especially in the economically developed regions. Spatially, the JJA mean MDA8 increased from south to north in eastern China (Figure 1a). To the south of 28oN (i.e., South China), the mean MDA8 was mostly lower than 100 μg/m³ and the ozone pollution was obviously lower than that in North China and in the Huanghuai area (NH). It is notable that MDA8 in the PRD was relatively higher than that for the surrounding

areas. The mean MDA8 was above 130 μg/m³ to the north of 32°N (i.e., the NH area), and thereinto, the large values of MDA8 centred on the Beijing-Tianjin-Hebei region and in western Shandong province exceeded 150 μg/m³. In the transitional zone, i.e., between 28°N and 32°N, the MDA8 varied from 100 μg/m³ to 120 μg/m³. Surface O3 pollution was closely linked to the anthropogenic emissions that dispersed and concentrated in the large cities. In the YRD and PRD, high levels of MDA8 were around the large cities. However, the high-level O3 values in the NH region were contiguous, indicating extensively severe

surface O3 pollution.

The maximum values of MDA8 for four summers were extracted to evaluate the levels of O₃ pollution from another angle (Figure 1b). To the north of 30°N, the maximum MDA8 at most sites was above 265μg/m³, indicating that the levels of O₃ pollution had exceeded the threshold of severe surface O₃ pollution in China (The Ministry of Environmental Protection of China, 2012). In Beijing and Tianjin, two large cities in NC, the MDA8 values were nearly above 100 μg/m³ and frequently



exceeded 215 μg/m³ (Figure 2a). The percentage of non-O3-polluted days (<100 μg/m³) and moderate O3-polluted days (>215

μg/m³) were 14.4% and 15.3%, respectively, indicating that more than 85% O$_3$ concentrations exceeded the health threshold

and that for more than 15% of summer days, O$_3$ concentrations moderately damaged human health. The maximum MDA8 in

the north of Hebei province and in eastern Shandong province even exceeded 320 μg/m³, which badly injured the health of

local citizens. In Shijiazhuang and Weifang, the MDA8 levels were lower than those in Beijing and Tianjin during 2015–2016,

but dramatically increased to levels comparable to those of Beijing and Tianjin in 2017 and 2018 (Figure 2b). In Nanjing and

Shanghai, located in the YRD, the MDA8 did not show a clear increasing trend (Figure 2c). Similar to the distribution of the

mean MDA8, the maximum MDA8 to the south of 30oN was lower by comparison. Although approximately 60% of summer

days were non-O$_3$-polluted in the cities of Guangzhou and Zhongshan (Figure 2d), severe O$_3$ pollution also occurred in the

PRD (Figure 1b). The surface O$_3$ levels in Fujian province were the lowest seen in eastern China, represented by both the mean

MDA8 of 70–90 μg/m³ and maximum MDA8 of 160–200 μg/m³. These features, i.e., high-level MDA8 and south-north

differences, can also be observed in the MDA8 measurements for each year (Figure S1).

For the ten representative stations with severe ozone pollution, the daily variations in MDA8 were evident (Figure 2–3),

indicating large influences by the daily meteorological variables. Generally, the peak in the summer surface ozone

concentrations occurred in June in the northeast China, and then decreased in July-August due to abundant rainfall in Beijing

(Ma et al., 2016). Except for 2015, monthly MDA8 peaked in June and then declined in July and August for the sites to the

north of 30ºN, i.e., in Beijing, Tianjin, Tangshan, Taiyuan, Weifang, Shijiazhuang (Figure 3a–b). However, similar monthly

peaks were not obvious for the cities to the north of 30ºN (Figure 3c–d).

Considering the characteristics of the observed MDA8 mentioned above, an empirical orthogonal function (EOF) was used to

explore the dominant patterns of summer ozone pollution in eastern China (Figure 4). The percentage of variance contribution

for the first three patterns were 21.5%, 15.5% and 8% respectively. Approximately 37% of the variability in the original data

was contained in the first two patterns, therefore, they were defined as the dominant patterns of surface ozone pollution. In the

first EOF pattern (PAT1), the observed MDA8 at different sites changed similarly and the centre of variation was located in the

NH area (Figure 4a). The time series of EOF1 showed that the ozone pollution during 2017–2018 was more serious than that in

2015 and 2016 (Figure 4b). Differently, the second EOF pattern (PAT2), showed notable south-north difference, with centres

115 in the NC and YRD regions (Figure 4c). The time coefficient of PAT2 also did not show an obvious increasing trend (Figure

4d). The PAT1P (PAT2P) and PAT1N (PAT2N) events were defined as when the time series of EOF1 and EOF2 were greater

than one standard deviation and less than － 1×one standard deviation, respectively. Figure 4 illustrates the composite results

for the dominant patterns of surface ozone. The ozone concentrations for the PAT1P classification were generally greater than

those for PAT1N (Figure 5a-b). The MDA8 values in the NH region were >160 μg/m³ and <120 μg/m³ for PAT1P and PAT1N,

120 respectively, indicating significant differences. For the second pattern, the PAT2P appeared as a diminishing pattern from the


north to the south, however, there was severe ozone pollution in the YRD under PAT2N conditions. Therefore, the centres of O₃ variation were NH for the PAT1, and NC and the YRD for the PAT2. From the variations in summer MDA8 (Figure 6), the surface O₃ concentrations in these three areas increased from 2015 to 2017 and remained high in 2018 (Figure 6).

## 4. Associated atmospheric circulations

125 Associated atmospheric circulations were composited for PAT1 (PAT1P minus PAT1N) and PAT2 (PAT2P minus PAT2N). For the first pattern, the most noticeable O₃-changed region was NH (Figure 4a). The correlation coefficient between the time series of PAT1 and the NH-averaged MDA8 was 0.97 (Table 1). Thus, the effects of the anomalous atmospheric circulations mainly acted on the photochemical reactions near the surface in NH. There were negative Z850 anomalies over the Ural Mountains, indicating a weaker high ridge. Over the broad region from eastern Eurasia to the north Pacific, the anomalous

130 atmospheric circulations were located zonally, i.e., positive Z850 on the subtropical zone, with cyclonic anomalies at the mid to high latitudes and positive anomalies on the polar region (Figure 7a). The East Asia deep trough was enhanced and extended to northeast China and Japan. The intensity of the East Asia deep trough (i.e., the negative area-averaged Z850) positively correlated with the time series of PAT1 (Table 1) with a correlation coefficient of 0.28 (above the 99% confidence level). In accordance with the deep positive height anomalies to the north of Lake Baikal, which also extended southward to the edge of

135 the Tibetan Plateau, cold air activity was transported to the lower latitudes, but did not arrive at the NH region (Figure 7a). Influenced by the enhanced East Asia deep trough, the main body of WPSH shifted southward. The location of WPSH ($Z500_{(125^0 E,\ 20^0 N)} - Z500_{(125^0 E,\ 30^0 N)}$ ) also showed a positive correlation with the time series of PAT1 (R=0.39, Table 1). However, the western ridge point of WPSH was northward and westward than normal (being indicated by $Z500_{(110^0 E,\ 30^0 N)}$), and occupied the NH area, which was significant with the time series of PAT1 (R=0.24, above the 99% confidence level).

140 Although the local anomalous anticyclone over the east of China seemingly delivered water vapor to North China (Figure 7b), the channel of moisture was already cut off in the ocean at low latitudes by the positive and zonal anomalies in the subtropical regions (Figure 7a) and resulted in a dry environment in NH from surface to 500 hPa (Figure 7c). Furthermore, the associated descending motions not only enhanced the efficient adiabatic heating (Figure 7c), but also suppressed the development of convective activity (Figure 7d). The correlation coefficients between the time series of PAT1 and NH-averaged precipitation,

145 SAT, and downward solar radiation at surface were –0.44, 0.14 and 0.45, respectively, all of which exceeded the 99% significance test (Table 1). The large-scale atmospheric circulations led to sunny days with high temperatures near the surface (Figure 7a), less precipitation (Figure 7b), a dry environment (Figure 7c) and intense solar radiation (Figure 7d), which substantially enhanced the generation of ozone in NH.

Large amplitudes of PAT2 O₃ were distributed in the NC and YRD regions, and furthermore, the correlation coefficient between the time series of PAT2 and the MDA8 difference between NC and the YRD was 0.77 (Table 1). The impacts of




atmospheric circulations on the photochemical reactions in the above two areas are analysed in Figure 9. Due to the broad positive Z500 anomalies at the high latitudes of Eurasia, the subjacent surface air temperatures significantly increased, indicating weak cold air activity (Figure 9a). Moreover, there were positive Z500 anomalies from the Chukchi Peninsula to Northeast China. In summer, anomalous anticyclonic circulations at the mid and high latitudes generally led to significantly

positive SAT anomalies (Figure 9a). The East Asia deep trough was stronger (R=0.3), but was limited to the east of Japan.

 Extruded by the East Asia deep trough and cyclonic anomalies from the Siberian plains to the YRD, the WPSH moved southward and exhibited southwest-northeast orientation (Figure 9a). The location of WPSH ( $Z500_{(110^0E,\ 20^0N)} -$ $Z500_{(110^0E,\ 30^0N)}$ ) was positively correlated with the time series of PAT2 (R=0.32, Table 1). The southwest-northeast distribution of WPSH aided water vapor transportation to the YRD region (Figure 9b–c). Combined with significant upward

air flow (Figure 9c), more clouds formed at the medium and low levels (Figure 9d) and precipitation was enhanced in the YRD region (Figure 9b). A moist-cool environment, weak solar radiation and obvious wet deposition reduced the ozone concentration in the YRD region. On the other hand, sinking motion with efficient adiabatic heating (Figure 9c) and weak cold activity (Figure 9a) from the north both resulted in a temperature increase in NC (Figure 9a). There was divergence of water vapor and less cloud cover over NC, resulting in dry, hot and sunny weather (Figure 9b, d). Under such meteorological

conditions, the generation of surface $O_3$ was accelerated, and thus, higher MDA8 was observed in NC. The differences in precipitation, SAT, and downward solar radiation at the surface between the NC and YRD regions were calculated and their correlation coefficients with the time series of PAT2 were –0.46, 0.18 and 0.62, respectively (Table 1). The significant correlations indicated that the differences in meteorological conditions between NC and YRD regions, associated with the aforementioned anomalous atmospheric circulations, largely contributed to $O_3$ PAT2.

**5.   Signals for interannual variability**

The O3 pollution in NH became persistently severe over the past four years, reflected in both the O3-polluted areas and in the O3 concentration (Figure 11). In addition, the number of sites with maximum MDA8 > 265 µg/m³, located in NH and the YRD, also increased year by year. The summer MDA8 in the PRD was not as high as that in in NH and the YRD (Figure 1a), but its maximum O3 concentration exceeded the threshold for severe surface O3 pollution in China in each year (Figure 11). The

observed summer MDA8 in eastern China also presented evident interannual differences (Figure 11). For example, the spatial patterns of MDA8 anomalies were different in each year (Figure S2). Although the relative variance contributions of the spatial coefficients varied, the first two EOF patterns of MDA8 were always PAT1 and PAT2 in different years, indicating that the extracted dominant patterns were reliable and steady. Sorting by the variance contribution, the dominant patterns were PAT2 and PAT1 in 2015 and 2016 (Figure S2a–d), however, they are PAT1 and PAT2 in the two subsequent years (Figure S2e–h).

A question raised here is whether the aforementioned composited signals of atmospheric circulations could provide



implications for the climate variability of the summer $O_3$ pollution in eastern China. In 2016 and 2018, the variance contribution of the first pattern was almost twice that of the second pattern, and thus, these two years were selected as typical years whose varied patterns were clearly separated. The dominant pattern of 2016 was PAT2 (explaining approximately 24% of the variance), while the MDA8 in 2018 changed as PAT1, with nearly 34% variance contributions (Figure 11d). In 2016, the

MDA8 values in NC and the YRD were nearly out of phase (Figure 12a), and the correlation coefficient between them was –0.28 (above the 99% confidence level). Differently, this correlation coefficient was 0.43 (Figure 12b), indicating similar change features in 2018 between the MDA8 levels of NC and the YRD.

The MDA8 anomalies in 2016 were negative in NC, but positive in the YRD and PRD (Figure 10b). The interannual anomalies of atmospheric circulations in 2016, with respect to the mean of 2105–2018 (Figure 13), were consistent with the associated

atmospheric circulations for PAT2 (Figure 9). There were positive Z500 anomalies over the north Pacific at the mid to high latitudes (Figure 13a). These positive anomalies not only indicated a weaker East Asia deep trough but also induced a shallow trough from the Chukchi Peninsula to Northeast China. Together with the stronger high ridge over the Ural Mountains, the cold air was transported to the mid latitudes, resulting in a lower SAT for North China (Figure 13a). The western segment of the WPSH was stronger and moved northward, which occupied the YRD and south China and brought moist air flows to North

China (Figure 13b). Sufficient moisture formed more low to mid-level clouds, causing a decrease in solar radiation reaching the ground in NC (Figure 13d). Associated with the extending WPSH, sinking motions resulted in a hot, dry air mass near the surface in the YRD region (Figure 13b–c). In addition, decreased cloud cover did not effectively reflect solar radiation, which was an essential condition for the enhancement of the photochemical reactions. Therefore, at the interannual time scale, the atmospheric anomalies, similar to those shown in Figure 9, also played important roles on the spatial pattern of MDA8

anomalies. That is, the atmospheric circulations in 2016 accelerated the formation of surface O3 in the YRD, but weakened the summer O3 pollution in NC.

The MDA8 anomalies were mostly positive in the east of China in 2018 (Figure 10d). "–+–" Z850 anomalies were located over the Ural Mountains and to the north of Lake Baikal and the Aleutian Islands (Figure 12a), which was consistent with the anomalous patterns in Figure 7. The East Asia deep trough shifted northward, and meanwhile, the western ridge point of

WPSH also shifted northward, resulting in a higher SAT in the east of China and accelerating the photochemical conversion for elevating the surface ozone concentration. The local anomalous anti-cyclone over the NH and the Japan Sea also existed in the interannual signals, which induced the divergence of water vapor in southeast China (Figure 12b). Due to the lack of moisture, it was difficult for cloud cover to form, and more solar radiation directly reached the ground (Figure 12d). The large-scale atmospheric circulations led to high temperatures near the surface in eastern China, and to a dry and sunny environment in the

YRD and South China. Thus, under such weather conditions, positive MDA8 anomalies were observed in 2018. Although the signals of global warming had impacts on the anomalies of atmospheric circulations, the key features could also be found when



the anomalies were viewed with respect to the climate mean during 1979–2018 (Figure S3–S4). In addition, if the variance contributions of the first two EOF patterns were not significantly different, i.e., in 2015 and 2017, the interannual signals of atmospheric circulations also showed integrated characteristics of PAT1 and PAT2 (Figure S5–S6).

**6.    Conclusions and discussions**

During the recent four years, ground-level ozone pollution became the major air challenge in the summers in the east of China. The highest $O_3$ concentrations were observed in North China and in the Huanghuai region, which are located north of 32ºN. The $O_3$-contaminated air occurred for 85% of summer days in Beijing and Tianjin. In the south, the surface $O_3$ pollution was also severe both in the Yangtze and Pearl River delta regions. Meteorological conditions had significant impacts on the evident

daily fluctuation of MDA8, to reveal their detailed relationships, the dominant patterns of summer ozone pollution and associated atmospheric circulations were analysed in this study.

The MDA8 of the first prominent pattern changed synergistically in the east of China, especially in North China and in the Huanghuai region. An enhanced east Asia deep trough and west Pacific subtropical high were zonally distributed and prevented the northward transportation of moisture. The northward-shifted western ridge point of the west Pacific subtropical

high accelerated the photochemical reactions via hot-dry air and intense solar radiation. The second pattern of ozone pollution showed remarkable south-north differences. Broad positive geopotential height anomalies at the high latitudes significantly increased the surface air temperature and thus decreased cold air activities. These positive anomalies also extended to North China and resulted in locally warmer air near the surface. On the other hand, the southwest-northeast oriented west Pacific subtropical high transported sufficient water vapor to the Yangtze River Delta. Consequently, a local moist-cool environment,

without intense sunlight, reduced the formation of surface ozone.

The interannual differences in summer O3 pollution were also discussed and the composited signals of atmospheric circulations and weather conditions proved to have meaningful implications for climate variability. Although the simultaneous large-scale atmospheric circulations were diagnosed on an interannual scale, their possible preceding climate drivers, e.g., sea ice, and sea surface temperature, were still unclear. The research related to climate variability has always needed long-term

data. To get around the problem of the data time span, Yin et al (2019) developed an ozone weather index using data from 1979 to 2017 and demonstrated the contributions of Arctic sea ice in May to $O_3$ pollution in North China. In addition, the dominant patterns of ozone concentrations were also decomposed with the observed data from 2015 to 2018. With the increase in $O_3$ observations, increasingly reliable dominant patterns and more features might be revealed in the future. At present, the fine particulate matter decreased in the summers in eastern China (Li et al. 2018); however ozone production was significantly

enhanced. Thus, attentions to surface pollution should be strengthened and the weather-climate component should be taken into account when making decisions for control measures.



*Author contribution*

ZY and HW designed the research. BC and ZY performed most of the Figures and analysis. ZY prepared the paper with

contributions from all co-authors.


*Competing interests*

The authors declare that they have no conflict of interest.

*Acknowledgements*

This research was supported by the National Natural Science Foundation of China (41421004, 91744311 and 41705058) and

the funding of the Jiangsu Innovation & Entrepreneurship team.

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

**Figures captions**

**Table 1.** Correlation coefficients between the time series of PAT1 (PAT2) and the key indices of atmospheric circulations and meteorological conditions. "**" and "*" indicate that the correlation coefficients were above the 99% and 95% confidence level, respectively.



**Figure 1.** Distribution of the (a) mean values and (b) maximum values of MDA8 in summer from 2015 to 2018. The black dots indicate the locations of the observation sites.

**Figure 2.** Variations in MDA8 of polluted cities from 2015 to 2018, including (a) Beijing, Tianjin and Tangshan; (b) Taiyuan, Weifang and Shijiazhuang; (c) Shanghai and Nanjing; and (d) Zhongshan and Guangzhou.

**Figure 3.** Boxplots of the MDA8 in June, July and August from 2015 to 2018. The polluted cities included (a) Beijing,
Tianjin and Tangshan; (b) Taiyuan, Weifang and Shijiazhuang; (c) Shanghai and Nanjing; and (d) Zhongshan and Guangzhou.

**Figure 4.** The first EOF pattern (PAT1: a, b) and second EOF pattern (PAT2: c, d) of MDA8 in summer from 2015 to 2018, including the spatial pattern (a, c) and the time coefficient (b, d). The black boxes in panels a and c are the selected North China and Huanghuai region (NH), North China (NC) Yangtze River Delta (YRD) and Pearl River Delta (YRD).

**Figure 5.** Composites of the MDA8 PAT1 (a, b) and PAT2 (c, d) in summer from 2015 to 2018. Panels a and c were composited when the time coefficient of EOF1 and EOF2 was greater than one standard deviation, while panels b and d were when the time coefficient was less than $-1 \times$ one standard deviation.

**Figure 6.** Variations in the area-mean summer MDA8 from 2015 to 2017 for NH, NC, and YRD.

**Figure 7.** Composites of the daytime atmospheric circulations (i.e., PAT1P – PAT1N). (a) Geopotential height at 850 hPa
(contours) and surface air temperature (shading), (b) water vapor flux at 850 hPa (arrows) and precipitation (shading), (c) 100°E–120°E mean wind (arrows) and relative humidity (shading), (d) downward solar radiation at the surface (shading) and the sum of low and medium cloud cover (contours). The white dots indicate that the shading was above the 95% confidence level. The purple boxes in panels a, b and d show the NH region, and the black box in panel a indicates the location of the East Asia trough.

**Figure 8.** Variations in standardized PAT1 time series, NH-area averaged MDA8 and the key indices of atmospheric circulations (a), and the meteorological conditions (b). The calculations of $EAT_1$, $WPSH_1$, $Pre_1$, $SAT_1$ and $SSR_1$ are consistent with those in Table 1.

**Figure 9.** Composites of the daytime atmospheric circulations (i.e., PAT2P – PAT2N). (a) Geopotential height at 500 hPa (contours) and surface air temperature (shading), (b) water vapor flux at 850 hPa (arrows) and precipitation (shading), (c)
100°E–120°E mean wind (arrows) and relative humidity (shading), (d) downward solar radiation at the surface (shading) and the sum of low and medium cloud cover (contours). The white dots indicate that the shading was above the 95% confidence level. The purple boxes in panel a, b and d are the NC and YRD regions, and the black box in panel a indicates the location of the East Asia trough.

**Figure 10.** Variations in the standardized PAT2 time series, the MDA8 difference between NC and the YRD and the key
indices of atmospheric circulations (a), and the meteorological conditions (b). The calculations of $EAT_2$, $WPSH_2$, $Pre_2$, $SAT_2$ and $SSR_2$ are consistent with those in Table 1.

**Figure 11.** Anomalies in the summer mean MDA8 in 2015 (a), 2016 (b), 2017 (c) and 2018(d), relative to the mean during 2015–2018. The black crosses and dots indicate the locations of the observation sites while the black crosses also indicate that the maximum MDA8 was larger than 265 μg/m³.

**Figure 12.** Variations in the MDA8 of NC (black) and the YRD (blue) in 2016 (a) and 2018 (b).

**Figure 13.** Anomalies of summer mean atmospheric circulations in 2016, with respect to the mean during 2015–2018. (a) Geopotential height at 500 hPa (contours) and surface air temperature (shading), (b) water vapor flux at 850 hPa (arrows) and precipitation (shading), (c) 100°E–120°E mean wind (arrows) and relative humidity (shading), (d) downward solar radiation at the surface (shading) and the sum of low and medium cloud cover (contours). The white dots indicate that the
shading was above the 95% confidence level.

**Figure 14.** Anomalies of summer mean atmospheric circulations in 2018 with respect to the mean during 2015–2018. (a) Geopotential height at 850 hPa (contours) and surface air temperature (shading), (b) water vapor flux at 850 hPa (arrows) and precipitation (shading), (c) 100°E–120°E mean wind (arrows) and relative humidity (shading), (d) downward solar radiation at the surface (shading) and the sum of low and medium cloud cover (contours). The white dots indicate that the
shading was above the 95% confidence level.





**Table 1.** Correlation coefficients between the time series of PAT1 (PAT2) and the key indices of atmospheric circulations and meteorological conditions. "**" and "*" indicate that the correlation coefficients were above the 99% and 95% confidence level, respectively.

| PAT1 | $MDA8_1$ | $EAT_1$ | $WPSH_1$ | $Pre_1$ | $SAT_1$ | $SSR_1$ |
|---|---|---|---|---|---|---|
|  | $0.97^{**}$ | $0.28^{**}$ | $0.39^{**}$ | $-0.44^{**}$ | $0.14^{**}$ | $0.65^{**}$ |
| PAT2 | $MDA8_2$ | $EAT_2$ | $WPSH_2$ | $Pre_2$ | $SAT_2$ | $SSR_2$ |
|  | $0.77^{**}$ | $0.30^{**}$ | $0.32^{**}$ | $-0.46^{**}$ | $0.18^{**}$ | $0.62^{**}$ |

$MDA8_1$ is the NH-area averaged MDA8, while the $MDA8_2$ is the MDA8 difference between NC and YRD. $EAT_1$ and $EAT_2$ indicate the intensity of the East Asia deep trough and were calculated as the mean $-Z850$, shown in the black boxes in Figure 7 and Figure 9, respectively. $WPSH_1$ ($Z500_{(125^0E,\ 20^0N)} - Z500_{(125^0E,\ 30^0N)}$) and $WPSH_2$ ($Z500_{(110^0E,\ 20^0N)} - Z500_{(110^0E,\ 30^0N)}$) represents the location of WPSH. $Pre_1$, $SAT_1$ and $SSR_1$ were calculated as the NH-area averaged precipitation, SAT and downward solar radiation at the surface, respectively. $Pre_2$, $SAT_2$ and $SSR_2$ are the differences in the NC- and YRD-area averaged precipitation, SAT and downward solar radiation at the surface, respectively.


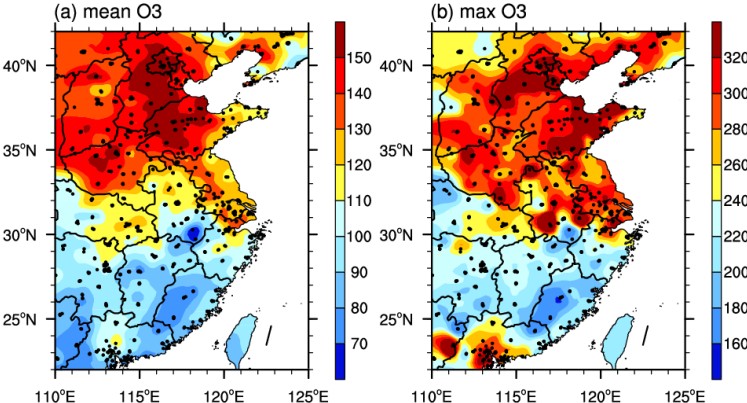

**Figure 1. Distribution of the (a) mean values and (b) maximum values of MDA8 in summer from 2015 to 2018. The black dots indicate the locations of the observation sites.**


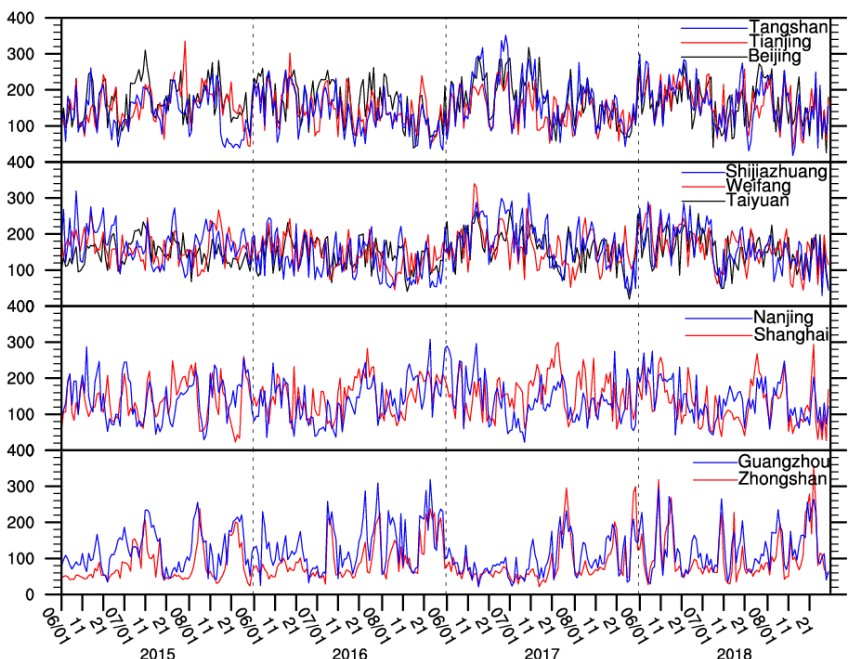

**Figure 2. Variations in MDA8 of polluted cities from 2015 to 2018, including (a) Beijing, Tianjin and Tangshan; (b) Taiyuan, Weifang and Shijiazhuang; (c) Shanghai and Nanjing; and (d) Zhongshan and Guangzhou.**

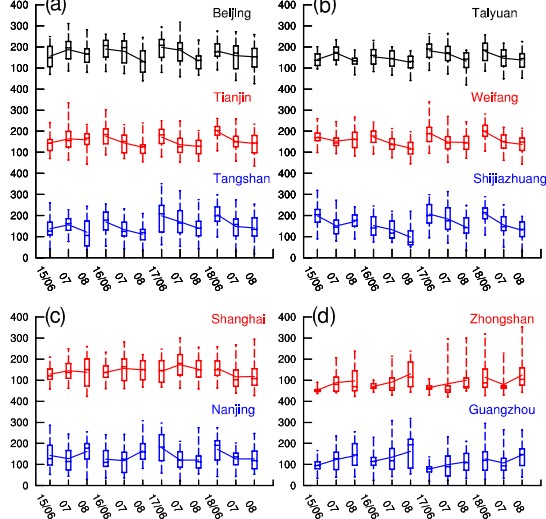

**Figure 3. Boxplots of the MDA8 in June, July and August from 2015 to 2018. The polluted cities included (a) Beijing, Tianjin and Tangshan; (b) Taiyuan, Weifang and Shijiazhuang; (c) Shanghai and Nanjing; and (d) Zhongshan and Guangzhou.**





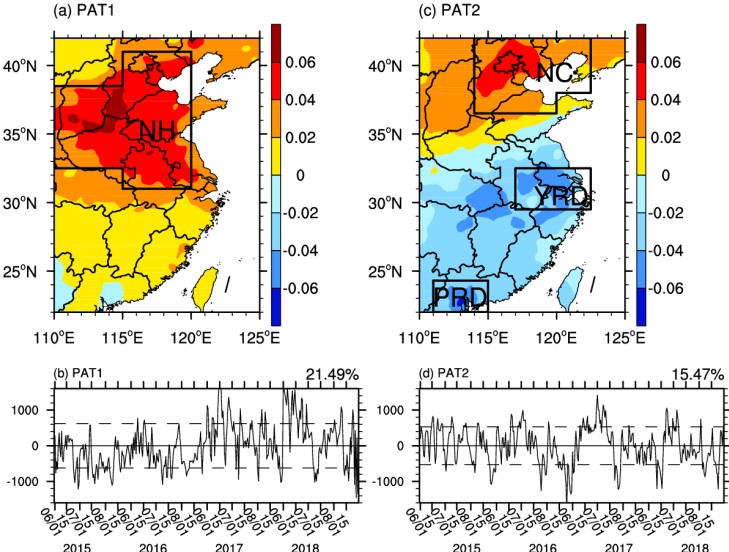

**Figure 4. The first EOF pattern (PAT1: a, b) and second EOF pattern (PAT2: c, d) of MDA8 in summer from 2015 to 2018, including the spatial pattern (a, c) and the time coefficient (b, d). The black boxes in panels a and c are the selected North China and Huanghuai region (NH), North China (NC) Yangtze River Delta (YRD) and Pearl River Delta (YRD).**

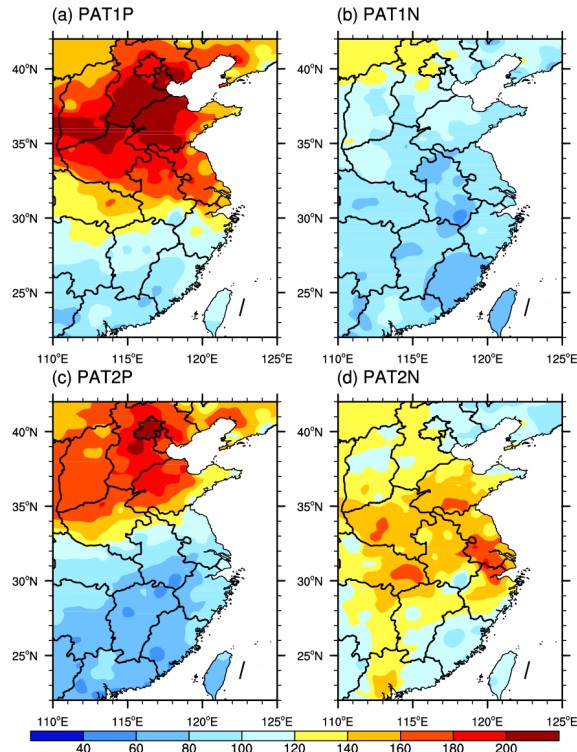

**Figure 5. Composites of the MDA8 PAT1 (a, b) and PAT2 (c, d) in summer from 2015 to 2018. Panels a and c were composited when the time coefficient of EOF1 and EOF2 was greater than one standard deviation, while panels b and d were when the time coefficient was less than − 1×one standard deviation.**

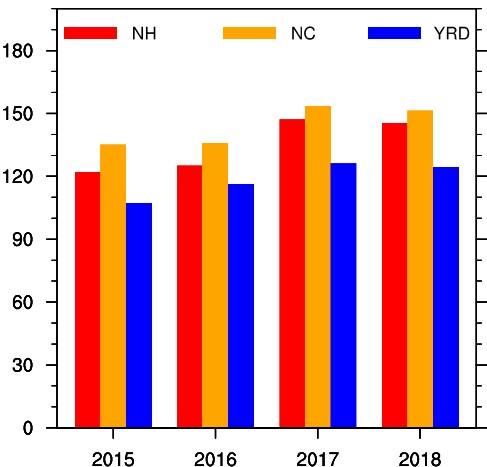

**Figure 6. Variations in the area-mean summer MDA8 from 2015 to 2017 for NH, NC, and YRD.**

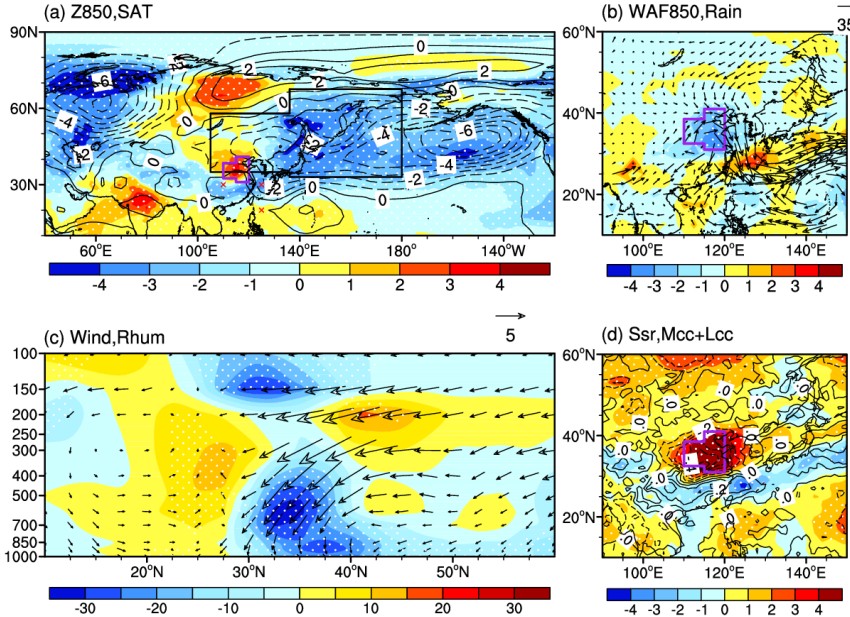


**Figure 7. Composites of the daytime atmospheric circulations (i.e., PAT1P – PAT1N). (a) Geopotential height at 850 hPa (contours) and surface air temperature (shading), (b) water vapor flux at 850 hPa (arrows) and precipitation (shading), (c) 100°E–120°E mean wind (arrows) and relative humidity (shading), (d) downward solar radiation at the surface (shading) and the sum of low and medium cloud cover (contours). The white dots indicate that the shading**
**was above the 95% confidence level. The purple boxes in panels a, b and d show the NH region, and the black box in panel a indicates the location of the East Asia trough.**


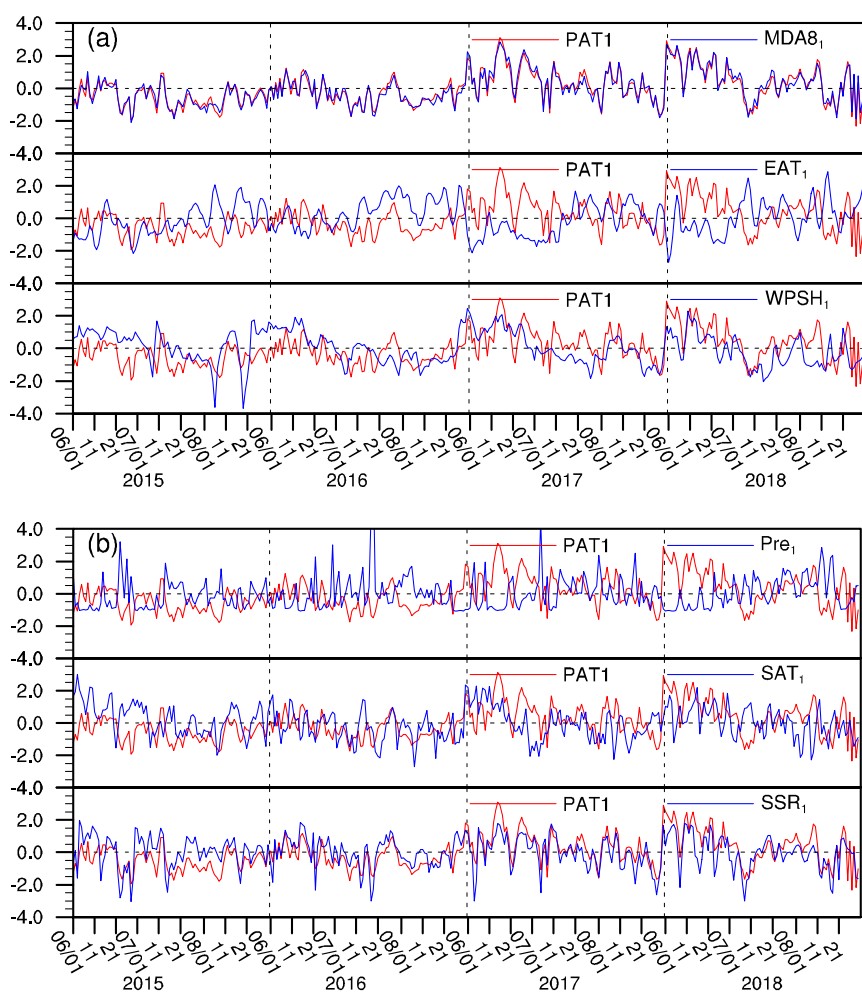

**Figure 8. Variations in standardized PAT1 time series, NH-area averaged MDA8 and the key indices of atmospheric circulations (a), and the meteorological conditions (b). The calculations of $EAT_1$, $WPSH_1$, $Pre_1$, $SAT_1$ and $SSR_1$ are consistent with those in Table 1.**





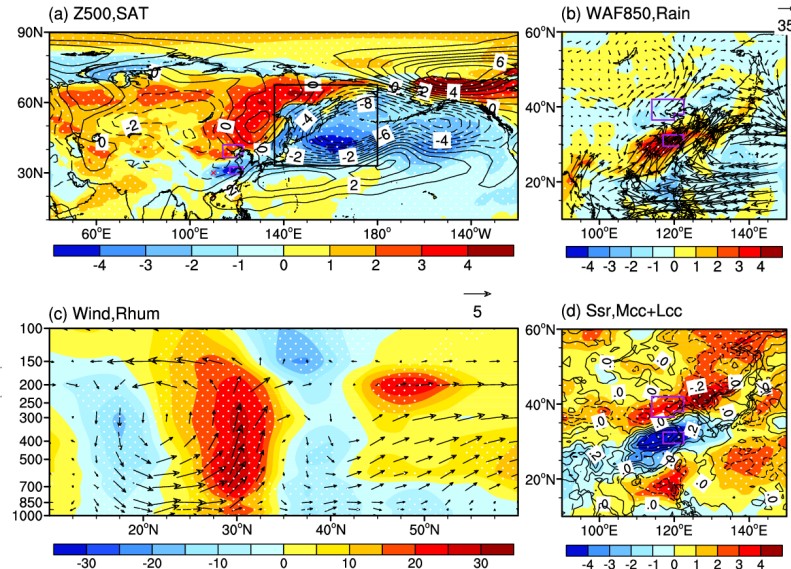


**Figure 9. Composites of the daytime atmospheric circulations (i.e., PAT2P – PAT2N). (a) Geopotential height at 500 hPa (contours) and surface air temperature (shading), (b) water vapor flux at 850 hPa (arrows) and precipitation (shading), (c) 100°E–120°E mean wind (arrows) and relative humidity (shading), (d) downward solar radiation at the surface (shading) and the sum of low and medium cloud cover (contours). The white dots indicate that the shading**

**was above the 95% confidence level. The purple boxes in panel a, b and d are the NC and YRD regions, and the black box in panel a indicates the location of the East Asia trough.**



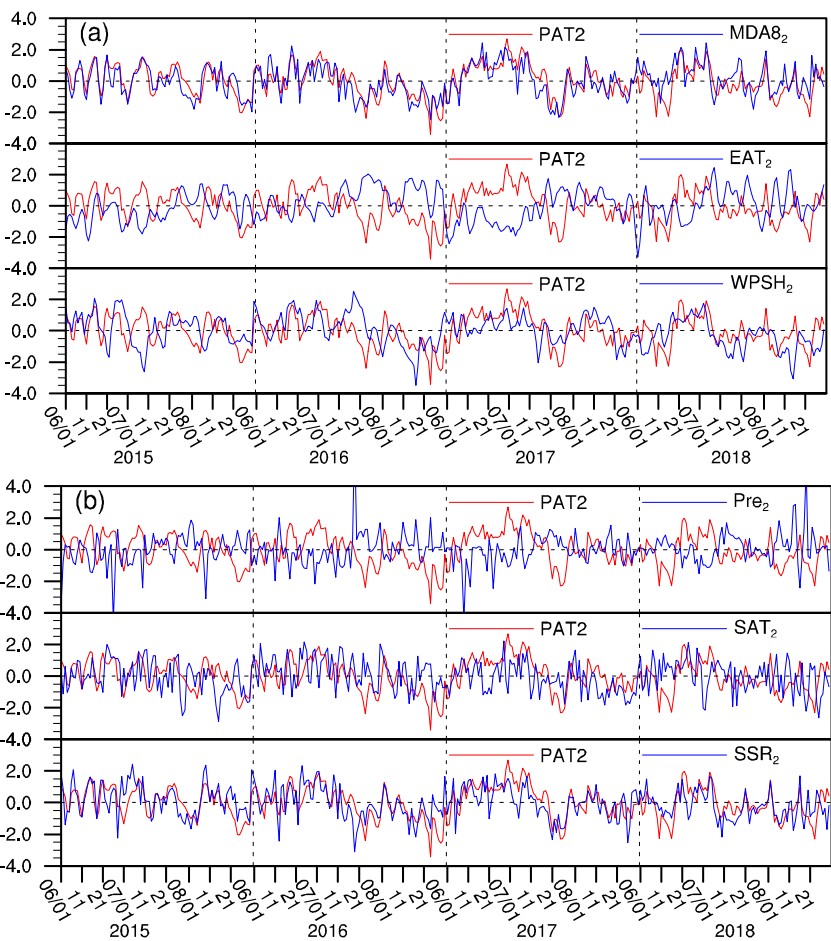

**Figure 10. Variations in the standardized PAT2 time series, the MDA8 difference between NC and the YRD and the key indices of atmospheric circulations (a), and the meteorological conditions (b). The calculations of $EAT_2$, $WPSH_2$, $Pre_2$, $SAT_2$ and $SSR_2$ are consistent with those in Table 1.**





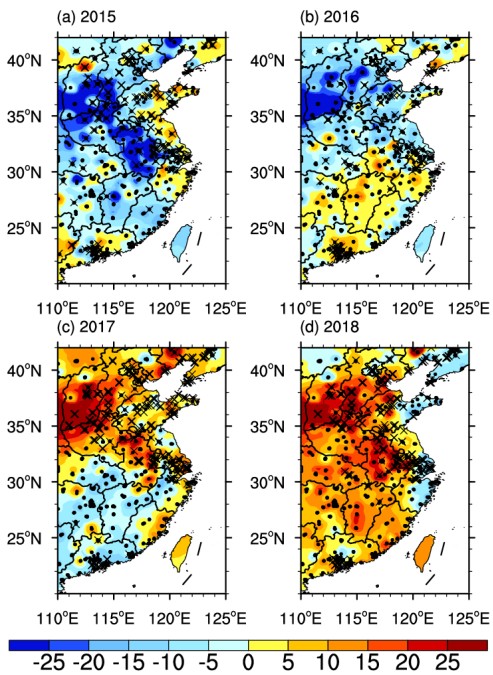

**Figure 11. Anomalies in the summer mean MDA8 in 2015 (a), 2016 (b), 2017 (c) and 2018(d), relative to the mean during 2015–2018. The black crosses and dots indicate the locations of the observation sites while the black crosses also indicate that the maximum MDA8 was larger than 265 µg/m³.**

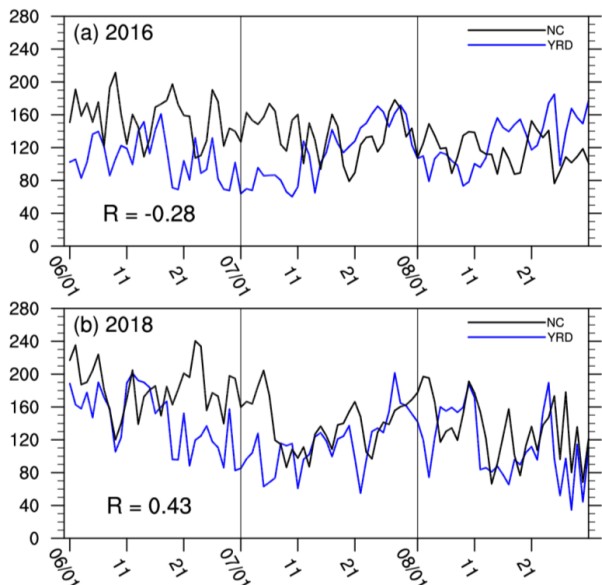

**Figure 12. Variations in the MDA8 of NC (black) and the YRD (blue) in 2016 (a) and 2018 (b).**

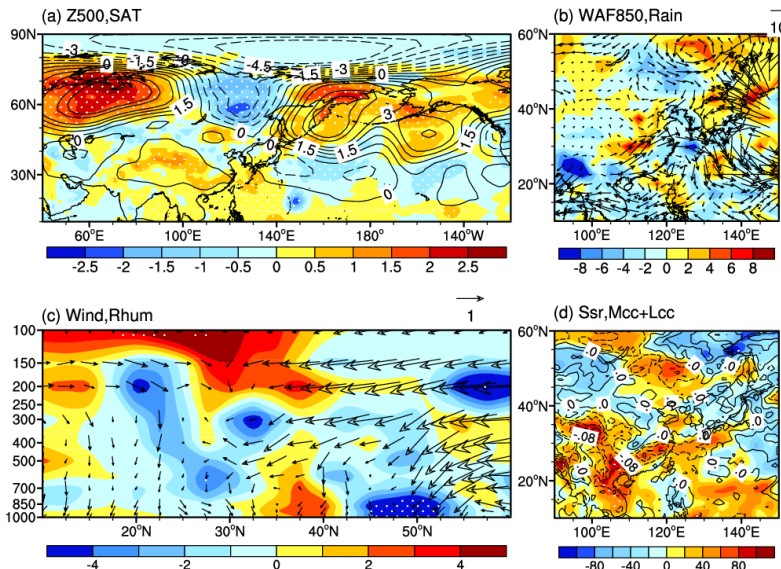

**Figure 13. Anomalies of summer mean atmospheric circulations in 2016, with respect to the mean during 2015–2018. (a) Geopotential height at 500 hPa (contours) and surface air temperature (shading), (b) water vapor flux at 850 hPa (arrows) and precipitation (shading), (c) 100°E–120°E mean wind (arrows) and relative humidity (shading), (d) downward solar radiation at the surface (shading) and the sum of low and medium cloud cover (contours). The white dots indicate that the shading was above the 95% confidence level.**

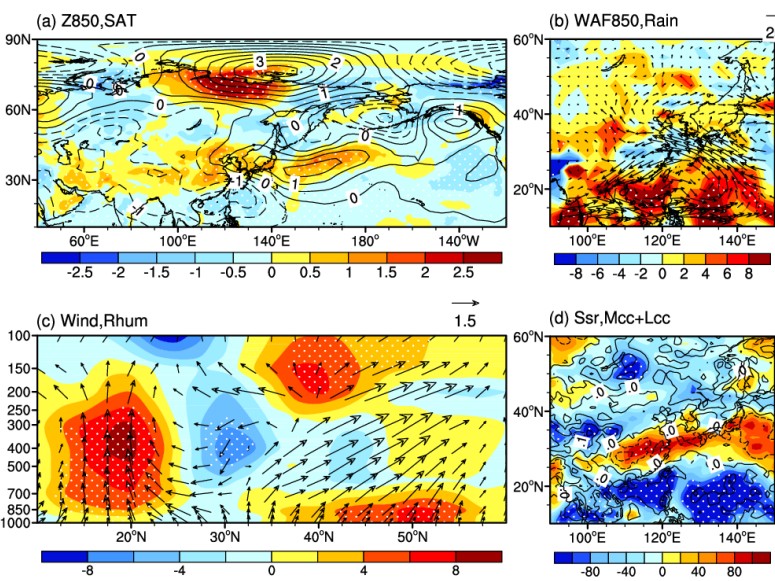

**Figure 14. Anomalies of summer mean atmospheric circulations in 2018 with respect to the mean during 2015–2018. (a) Geopotential height at 850 hPa (contours) and surface air temperature (shading), (b) water vapor flux at 850 hPa (arrows) and precipitation (shading), (c) 100°E–120°E mean wind (arrows) and relative humidity (shading), (d) downward solar radiation at the surface (shading) and the sum of low and medium cloud cover (contours). The white dots indicate that the shading was above the 95% confidence level.**