# Peer review of "Dominant Patterns of Summer Ozone Pollution in Eastern China and Associated Atmospheric Circulations"

_Atmospheric Chemistry and Physics, 2019_

## Referee Comment (RC1) · Anonymous Referee #1 · 21 Jul 2019

ACPD Review of "Dominant Patterns of Summer Ozone Pollution in Eastern China and Associated Atmospheric Circulations" By Zhicong Yin, Bufan Cao, and Huijun Wang

Summary:
This paper uses a combination of observation and reanalysis data to investigate the possible impact of large-scale meteorological conditions on surface air quality -- specifically ozone -- in eastern China.  Empirical Orthogonal Function (EOF) analysis of summertime daytime meteorology identified two EOF patterns which explained over a third of the variance for the 2015-2018 period.  The major drivers for high ozone in Eastern China were the location of the East Asia deep trough and the West Pacific Subtropical High, modulating favorable or unfavorable conditions for the formation of ground-level ozone.

I am not convinced that this study was ready for submission to ACP or exceptionally novel.  This work looked to be a continuation of the research published by Zhao and Wang (2017) (referenced on Line 51-53) who identified the link between the WPSH and high ozone in eastern China using observations from 2014-2016, with a focus on 2014.

The flow of this ACPD paper was at times a real struggle to read, with incorrect figure references, figures included which were never referenced, and statements about meteorology which were either wrong or difficult to interpret from the figures.  If there have been only a few years of ozone surface data to perform studies, I imagine there is not enough data to ascertain long-standing relationships (Line 57-59).  The authors need to strengthen the paper to sell this idea to their readers.

I recommend the manuscript undergoes major revisions.

Comments:

The authors should discuss the air quality thresholds of MDA8 O3 either in the Introduction or prior to listing all the values in Section 3 (Line 80-83).  It is not mentioned until lines 87-88, and 90.

It is not clear what was the methodology used to interpolate the station-based observations into the surrounding areas to fill in the maps in Figure 1 and the rest of the paper.  How many station sites are urban vs rural?  How did the number of sites change from year to year?  Regarding lines 83-85 starting with "In the YRD and PRD, high levels of MDA8 were around the large cities. However, the high-level O3 values in the NH region were contiguous, indicating extensively severe surface $O_3$ pollution." Was this expected or does it have anything to do with location and number of monitoring stations, or the algorithm to weight the station data into the surrounding area they each represent.

There is a disconnect between the Figure 1 and the text on Line 87 where 265 $\mu g/m^3$ is referenced but it is not a value on the color bar. I could support changing this to 240 $\mu g/m^3$.  If

you want to highlight 265 μg/m$^3$, consider changing the intervals on the color bar to 15 μg/m$^3$, starting at 175 μg/m$^3$ instead of 140 μg/m$^3$.

There must have been a more scientific reason for making Figure 2 other than "from another angle" (Line 86).  Are these stations close to each other?  Alternative reasons may be the authors wanted to investigate how well the cities represented an area over time, or maybe seasonal cycles or the year-to-year variability not captured in Figure 1a or 1b.  The authors should introduce this idea of the ten cities chosen as "representative stations with severe ozone pollution" (Line 102) earlier and describe what they are representative of.  Is it the region or province or something else?  Are they the top highest 2 or 3 cities in that region statistically or most populated in that region?

Any hypothesis why the seasonality in south china was different in summer of 2015 (Fig 3, Lines 105-107)?  Did it have anything to do with the El Nino that year? This idea was not revisited later in the paper with the analysis of the individual seasons EOF patterns.

The authors need to reconsider the figures submitted with this paper. Two major comments below are brought up here plus many minor comments at the end of my review:
1. It is unclear to me what Figure 6 is showing by "variations" (Line 122-123, 412).  There are no units on the right-hand side.  Is this area-averaged O3 for the different boxed regions (as outlined in Figure 4) for each summer?  Could it be better summarized in a table instead of a histogram, or maybe a line plot would take up less space?  There is also only the one sentence in the manuscript that references Figure 6.  Does it really show additional information not shown in another figure to keep it in the paper?
2. Figures 8 and 10 are not (correctly) referenced once in the text and therefore should be removed (or moved to the supplemental) and yet there are four lines of text referencing Figure S2 in the main text (Lines 175-179) and several lines of text referencing Figures S3-S6 (Lines 210-214).  The authors should also add references to Figure S2 in the discussion of the variance explained and the percentages given on Lines 181-184 which means that there is a decent amount of discussion of Figure S2 in the results section and lead to Section 5.  The authors should consider switching Figures 8 and 10 with Figure S2 in the supplemental material.  Can the authors comment more on the switch from PAT2 being the dominant EOF pattern from Summer 2016 to PAT1 being the dominant EOF in Summer 2017 (Line 179).

Minor and Technical Comments:
Line 23: 1) ozone at surface may come from the stratosphere through stratosphere to troposphere transport and 2) ozone is a secondary pollutant; it is not directly emitted. Declaring it is a 'man-made pollutant' is misleading.
Line 24:  need a space after "(Day et al., 2017)"
Line 28:  It would be helpful to reference the boxed regions for NC, YRD, and PRD are shown in Figure 4 (n.b., there is an error in Figure 4 caption (Line 407) where YRD is used twice when PRD should be used the second time).

Line 30 (and in several instances throughout the paper): the 3 in O3 needs to be a subscript

Line 32: Is this ppb short for ppbv or ppbm? Please change appropriately.

Line 33: a space is required between the value and the unit, here "2.1 ppbv". There are other instances like this (e.g., Line 87)

Line 42: Can the authors discuss in more detail the findings of the Li et al paper regarding the relationship between PM and O3 pollution.

Line 50: This sentence "Large-scale descending motion, tropical cyclones….related to the evaluation of surface O3" does not make sense to me as large-scale descending motion is usually associated with high pressure systems, not low-pressure systems.

Lines 54, 70: period after "et al."

Line 55-56: I have noticed also several papers coming out on this subject.  Can the authors provide references here of example studies.

Line 61, page 3: instead of "on the website" maybe say "publicly available" assuming this is true.

Line 68:  I would like to see more information on the reanalysis dataset.  2.5x2.5 is not the native resolution (nominally 0.7 degree) but the authors may have downloaded the data from ECMWF at this coarser resolution or degraded it themselves.

Line 69: What range and how many pressure levels?

Line 70-72: What is the temporal resolution of the original data (Hourly? 3-hourly? 6-hourly?) before it was selected to be "sub-daily".  What time in UTC is equivalent to 8am-8pm Beijing time.

Line 77: Superscript the circle to have a degree symbol.  Check throughout paper for this (e.g., line 97)

Line 78:  Why have the authors switched to using $\mu g/m^3$ from ppb in the Intro?

Line 79:  NH is a common abbreviation for Northern Hemisphere.  Does it stand here for **N**orth China and **H**uanghuai?  I suggest changing it to NCH, so as not to be confusing.

Line 80: It looks more to me that the cut off at 32°N is closer to $> 110 \mu g/m^3$ mean $O_3$ with much of the region $> 130 \mu g/m^3$, but not $> 130 \mu g/m^3$ everywhere.

Line 83: How do the authors know this "Surface O3 pollution was closely linked to the anthropogenic emissions that dispersed and concentrated in the large cities**"**? Reference to another study or figure from this text?

Line 89: Why is Tangshan not mentioned in this sentence along-side Beijing and Tianjin if it is included in the same Figure 2a panel?

Line 90-92: Two percentages are given, 14.4% and 15.3%, but this seems insufficient.  In the previous sentence the authors quote two cities, Beijing and Tianjin, frequently exceed moderate pollution levels, but in this sentence the authors refer to both non-polluted and moderate pollution days.  Are these percentages for the two cities for frequency of pollution days, or for one city and the percentage of non-polluted and polluted days?

Line 93: Which figure is this quoted from "the north of Hebei province and in eastern Shandong province even exceeded 320 $\mu g/m^3$,"?  Provide figure reference (I presume top panel of Figure 2 for Tangshan July 2018).

Line 94: Which region are the cities in Figure 2b (Shijiazhuang and Weifang).  Why do the authors not reference the third city from this panel, Taiyuan, in the text here?

Line 99-100: Where is the Fujian province?  This isn't labelled anywhere on a map and only referenced the once here in the paper, yet values are quoted.  Are these values form Figure 1 or Figure 2?

Line 103: Is there any reason we cannot assume some daily variation in emissions to impact variability in ozone, or is it all down to meteorology?

Line 104: Abundant rainfall in Beijing only, or should it be more general "in South China" to have impact on all 6 stations in Fig 3a,b?  Or is this sentence limited because the study referenced only looked at Beijing?  Can the authors explain how that idea can be expanded to other cities in the north of China?

Line 105:  Can the authors give a reason why 2015 was different for the northern cities?

Line 106-107: I presume the second reference to "cities north of 30" should actually be "cities south of 30".  Can the authors provide a reason(s) why the southern cities show less consistency in the month-to-month variability?

Line 116: The use of brackets around PAT2P and PAT2N is confusing here.  I suggest changing the start of the sentence to something like "The positive and negative phase for PAT1 and PAT2 are defined by the events that are greater than one standard deviation and less than -1*standard deviation, respectively (Figures 4b and 4d)."

Line 117-118: The placement of the statement starting with "Figure 4 illustrates the composite results…" seems out of place here, and then the results of Figure 5 are buried at the end of this paragraph defining the PAT1 and PAT2.  I suggest starting a new paragraph here but add to that sentence the connection to Figure 5: "Figure 4 illustrates the composite results …. surface ozone while Figure 5 shows the break down into the positive and negative phase composites."

Line 119: I suggest adding figure references to Fig 5a and Fig 5b following the numbers so the readers know which panel to look at.

Line 120: Have the authors performed a statistical analysis to claim the difference is significant?

Line 122: One could argue that there is higher O3 in the PRD during PAT2 as well.

Line 125: The authors showed ozone composites of PAT1 and PAT2 in Figure 4, so these figures are not the same and it should be clarified, not just by showing the defined differences in the parentheses.   I suggest changing the opening sentence here to "Anomalous atmospheric circulation associated with PAT1 (PAT1P composite minus PAT1N composite) and PAT2 (PAT2P composite minus PAT2N composite) are shown in Figures 7-10."  The Figure 7 caption (Line 416) says "Composites of the daytime atmospheric circulations" but it would be good to remind the reader in the main text here that these are daytime only and reference Section 2.

Line 126: Do the authors mean by "most O3-changed region" that the PAT1 positive and negative phases are defined by large differences in O3 within the NH region.  If so, I would change the sentence to something like my suggestion and reference Figure 5a,b instead of Fig 4.

Line 128-131: The authors need to be careful to get all their positive and negative descriptions correctly matching Figure 7a: e.g., Negative Z850 anomalies would indicate lower geopotential heights or a trough, not necessarily a ridge; there is not positive Z850 but negative across the subtropical Pacific, to get the cyclonic flow pattern.

Line 134: Can the authors provide approximate coordinates of Lake Baikal to help the reader identify the lake drawn at approximately 110E, 50N. I did not notice the lake at first; I thought it was a miscellaneous contour.

Line 136: WPSH shifted southward compared to what? Do I understand correctly further south during the PAT1N than PAT1P.

Line 137-139: reference heights are Z at 500 hPa but Figure 7a shows Z at 850 hPa. How can the authors say that the extent of WPSH is "indicated by Z500" if not shown anywhere and the reference to an R=0.24 is not in the Table 1 for PAT1.

Line 140: Start a new paragraph with "Although…." Hard to tell if the authors intend this to be a new paragraph since paragraphs do not start indented from the left margin.

Line 142: I read Fig 7c to be statistically drier from the surface to 400 hPa, not just up to 500 hPa.

Line 143-144: The authors should not put the reference for 7c after adiabatic heating since this is not shown in the figure and it is very difficult to ready the contours on 7d to know anything about convective activity, unless they are referring to the higher downward solar radiation at the surface only, and if that is true, that should be made clearer.

Line 146: Remove 'sunny'

Line 149: Add figure references (maybe Fig 4c or Fig 5c,d) for the statement "Large amplitudes of PAT2 O3 were distributed in the NC and YRD regions". I ask for this clarification as I do not understand what the authors mean by 'large amplitudes'; I can imagine the authors referring to higher ozone in NC during PAT2P while it is low in YRD and opposite pattern for PAT2N as shown in Figure 5c,d, but this needs to be made clear.

Line 152: Why are the authors now showing Z500 in Fig 9a when they showed Z850 in Fig 7a? Same question applies for Figures 13a and 14a.

Line 156: Is "Extruded" supposed to start a new paragraph or why is it slightly indented from the left margin?

Line 162-163: I do not understand the term "weak cold activity". Are the authors referring to less cold air advection from the north?

Line 172: Figure 11 is referenced before Figure 10 in the manuscript. The figures should be renumbered to match the order they are presented in the paper. **NOTE, after reading further it looks to be that Figure 10 was incorrectly referenced and this is part of a bigger issue regarding which figures should be included in the main text. **

Line 173: Can the authors quantify the change in the number of stations with polluted levels of MDA8, from how many in 2015 to how many in 2018? Also, can the authors clarify they are referring to the mean summer MDA8 O3 in the sentence "The summer MDA8 in the PRD….(Figure 1a)" since they reference Fig 1a.

Line 174: Not all sites in the PRD had max MDA8 O3 greater than the threshold in all four years (Fig 11). The authors should clarify this point.

Line 175: Is "eastern China" referring to the YRD region or the full region 110-125E, 22-42N?

Line 184: I believe this sentence incorrectly references Fig 11d and it should reference Fig S2.
Line 188: I believe this sentence incorrectly references Fig 10b and it should reference Fig 11b.
Line 189: change 2105 to 2015.

Line 190 and 199: The anomalous pattern in Figure 13a looks to me to be the **opposite** of Figure 9a, so do I interpret this correctly that 2016 is predominately in the negative phase of the PAT2. If that is correct, it should be discussed as such.
Line 194: I am having a hard time following the authors description without the purple boxes to indicate on the map the NH, NC, and YRD, e.g., I do not see the WPSH is near the YRD as described.

Line 202: I believe this sentence incorrectly references Fig 10d and it should reference Fig 11d.

Line 202-205: I am struggling with the description of the negative, positive, negative pattern in the Z850 and the link to the Figure 7. Also, is the description of the shift in the East Asia deep trough and the WSPH "northward" relative to Figure 7 or to something else? The authors could include again a black box to indicate the location of the East Asia deep trough similar to Figure 7 and 9 and also the coordinates or a box to indicate the WSPH.

Line 207-208: I believe this sentence incorrectly references Fig 12 when it should be Figure 14b and 14d, respectively.

Line 227: Again, I do not understand what is meant by "cold air activities".

FIGURE specific comments:

Line 89: Consider labeling or marking the cities of interest (e.g., Beijing and Tianjin; actually, all cities from Figure 2) on Figure 1a so it is easier for readers less familiar with Chinese geography. Another idea would be to provide a map next to Figure 2 time series with the cities labelled, possibly within the boxed regions shown in Figure 3.

Line 90: It might be too cluttered to do this; but, adding dashed lines at 215 $\mu g/m^3$ and 100 $\mu g/m^3$ on Figure 2 could help the reader to see the frequency of O3 concentrations above the moderate polluted level and below the non-polluted level.

Line 330, 395: There are no units in the Figure 1 caption nor on the Figure itself by the color bar. From the text on Page 3, I presume $\mu g/m^3$. I suggest removing the y-axis latitude labels of Figure 1b, since the same as Figure 1a, to reduce the amount of text between panels.

Line 332,397: Label each panel a) b) c) d) in Figure 2 to match the figure caption. Should there be a 'g' at the end of Tianjin as the line label shows in Figure 2a? Flip the order of the legend labels (i.e., Beijing on the top of the list, not the bottom) to match the order listed in the figure caption and to match the stacking of the box and whisker plots in Figure 3. Also, add labels

maybe to the left of each panel, indicating which region each panel represents.  If I followed the paper correctly, a) NC, c) YRD, and d) PRD, but from the text I wasn't sure where the cities in panel b come from, but only one left is NH.

Line 406: missing comma (NC)

Line 415: What are the red crosses on Figure 7a.  They are not mentioned in the figure caption.

Line 416, 436: Figures 7 and 8 are not simply a composite, but a difference of the composites, if I understand the "(i.e., PAT1P-PAT1N)" correctly.  I also find the use of blue/red color bar counter-intuitive to show drier air as blue and more precipitation/moist air as red in Fig 7b and c.  I suggest using either different color bar colors (such as brown/blue or brown/green) or flip the blue/red so red is for negative and blue for positive.

Line 430, 455: Figures 8 and 10 legends are hard to see as there is not much white space between the legend lines and the contours and they are close to the dashed lines separating the years.  Can they be added to the side, outside the panels?

Line 440: It is near impossible to see the purple boxes indicating NC and YRD regions, especially in panels a and d, when I printed the figures.  It is also hard to read the arrows in Fig 9b in the region of NC and where the arrows are long and dense over China and Japan.

Line 459: missing space "2018 (d)"

Line 465, 472: Are the anomalies in Figure 13 and Figure 14 also "daytime".  If so, include like in Figures 7 and 9.

---

## Referee Comment (RC2) · Anonymous Referee #2 · 1 Aug 2019

Summary This manuscript aims to analyze the dominant patterns of summer ozone over China in recent years, and associated circulations. While the topic is of importance to the field, the conclusions drawn from the study are not convincing to me. The methods in general lack novelty. The general patterns of ozone pollution and the association with meteorology have been reported in previous papers. The authors need to clarify the novelty and scientific contribution of this study. The presentation of this paper is also confusing. I had a hard time following the manuscript. The authors presented 14 figures, but most of them are quite confusing without clear explanations. There are a number of issues that should be addressed in order to make this paper suitable for publication. I have the following major comments and some minor comments.

Major Comments: 1. The spatial and temporal patterns of ozone could also be driven by anthropogenic emissions. The manuscript gave me an impression that ozone pattern in China is purely driven by circulation, which is not true. It's possible that the North-South pattern is mainly driven by emission variations. Also, the inter-annual variability in ozone may also be related to the emission changes in past years. The authors need to discuss how emission variations would affect their analysis.

2. The authors use ground-based observations of ozone describe the general patterns of ozone, but the distribution of ground-based sites is uneven. Most sites in China are urban sites, and there are few rural sites. There is no information how the authors infer spatial distribution of ozone (i.e. Figure 1) from limited sites.

3. The study relies on EOF analysis, but there is almost no explanations of how the EOFs are constructed, and why the first two patterns are indicative of the dominant patterns of ozone pollution. Only 37% variance can be explained with the first two EOFs ($\sim$ 20% for the first EOF), which is even less than half. I think it's necessary to explain the limitation of this statistical approach.

4. The authors included a lot of figures, but some seem to be redundant. For example, Figures 2 and 3 seem to be repetitive. Most of the figures are not very clear, yet the authors only spend one or two sentences explaining these figures. None of these figures is labeled clearly. There is even no unit for the numbers presented, which is unacceptable to me. I'd recommend the authors only keep those most important figures (e.g Figures 7 and 8), and expand their discussions on these figures.

5. Overall, the language of the manuscript should be further polished. There are several grammatical errors, which should be edited carefully.

Specific comments:

1. Line 70: It's not clear what 'sub-daily' means here. If it's four-hour data, which composites did you select?

2. Please double check the subscripts and superscripts of units and chemical names.

3. Lines 95 - 100: How did you calculate ozone levels in each province? Are the ground-based measurements spatially representative?

4. Line 125: How did you compose atmospheric circulations? This is an important step, but there is almost no explanation of the method.

5. Line 189: 2105 -> 2015

6. Line 200: The conclusion that atmospheric circulation accelerated ozone formation in YRD but weaken in NC is interesting, but this does not agree with ground-based observations, which do not show any enhancement of ozone in YRD in 2016 (nor decreased ozone in NC, Figure 6).

7. Line 205 - 201: The figure numbers are wrong?

8. Line 210: How did you draw the conclusion that positive MDA8 anomalies are observed in 2018? This conclusion seems to be inconsistent with Figure 6.

9. Figure 1, 7, 11: missing units.

10. Figures 2, 3, 8, 12: Missing y label.

11. Figure 4: The authors need to explain how they construct spatial and temporal EOFs, and what the figures show here. What do the numbers represent?

12. Figure 5: It's not clear why it is necessary to composite to positive and negative patterns. How does this help explain the results?

13. Line 202: Where is Figure 10d?

14. Figure 14 not referenced in the manuscript.

---

## Author Comment (AC1) · 8 Sep 2019

**Response to Reviewer #1**

Summary: This paper uses a combination of observation and reanalysis data to investigate the possible impact of large-scale meteorological conditions on surface air quality -- specifically ozone -- in eastern China. Empirical Orthogonal Function (EOF) analysis of summertime daytime meteorology identified two EOF patterns which explained over a third of the variance for the 2015-2018 period. The major drivers for high ozone in Eastern China were the location of the East Asia deep trough and the West Pacific Subtropical High, modulating favorable or unfavorable conditions for the formation of ground-level ozone.

1. I am not convinced that this study was ready for submission to ACP or exceptionally novel. This work looked to be a continuation of the research published by Zhao and Wang (2017) (referenced on Line 51-53) who identified the link between the WPSH and high ozone in eastern China using observations from 2014-2016, with a focus on 2014.

**Reply:**

Here, we explained the novelty of this study, particularly the differences with Zhao and Wang (2017), in the following four points. Also, we emphasized this novelty in the revised version.

(1) We revealed two dominant patterns, their varying sorts in different years and their associated anomalous atmospheric circulations. Although the north-south differential pattern was the first mode in 2014 (Zhao and Wang 2017), 2015 and 2016, it was sorted in the second place in 2017 and 2018 (Figure S5 in the revised manuscript). That is, our study not only revealed the two dominant patterns, what is more important, also **showed the varying features of the dominant patterns**. In the recent two years, the most dominant pattern was different from that in previous years, which is **a new feature and might related to the climate status**. Additionally, the comprehensive atmospheric circulations were analyzed, including the **location** of west Pacific subtropical high (WPSH), the East Asia deep trough and other atmospheric anomalies. In Wang and Zhao (2017), they solely focused on the impacts of the WPSH, particularly on the **accumulative enhancement** of WPSH.

(2) We clearly explained the anomalous atmospheric circulations related the  $O_3$  pollution **both in North China and in South China**. However, in Wang and Zhao (2017), the **physical mechanisms to impact O3 in North China was still not sufficiently explained** (referring to the weak correlation coefficients in the *green*)

*boxes* in Figure R1 d–f). We speculated the reason for insufficient explanations on O3 conditions in North China might be that the impacts form the mid-high latitudes were significant which was not involved in Zhao and Wang (2017). In our study, we found **both of the WPSH and the East Asia deep trough** had impacts on the O3 concentrations in the east of China (Table 1). Furthermore, we **paid more attentions to the O3 concentrations in North China** where the surface O3 polluted levels were much higher than in the Yangtze River Delta and Pearl River Delta. The **WPSH and East Asia deep trough jointly modulated** the local meteorological conditions to influence the O3 concentrations.

**Figure R1.** The Figure 6 in Zhao and Wang (2017). The summer mean fields of meteorological parameters (a–c) and their correlations with daily WPSH: total cloud cover (a, d), UV radiation (b, e) and near surface air temperature (c, f). The added green boxes indicate the location of North China.

**Table 1.** Correlation coefficients between the time series of PAT1 (PAT2) and the key indices of atmospheric circulations and meteorological conditions. "\*\*" and "\*" indicate that the correlation coefficients were above the 99% and 95% confidence level, respectively.

| PAT1 | MDA8 1 | EAT 1 | WPSH 1 |
|------|-------------------|------------------|-------------------|
|      | $0.97^{**}$       | $0.28^{**}$      | 0.39**            |
|      |                   |                  |                   |
| PAT2 | MDA82             | EAT 2 | WPSH 2 |

MDA81 is the NCH-area averaged MDA8, while the MDA82 is the MDA8 difference between NC and YRD. EAT1 and EAT2 indicate the intensity of the East Asia deep trough and were calculated as the mean -Z850, shown in the black boxes in Figure 7 and Figure 9, respectively. WPSH1 ( $Z500_{(125^{0}E, 20^{0}N)} - Z500_{(125^{0}E, 30^{0}N)}$ ) and WPSH2 ( $Z500_{(110^{0}E, 20^{0}N)} - Z500_{(110^{0}E, 30^{0}N)}$ ) represents the location of WPSH.

(3) The number and distribution of the sites are more sufficient and updated. In the EOF analysis of Zhao and Wang (2017), the number of O3 sties was only 191 in 2014 even in a larger study region than ours (Figure R2). The number (fewer than 200) and distributions (uneven) of the sites were limited, due to the establishment progress of the observation sites of atmospheric components in 2014.

Fig. 3. The EOF1 of daily summer MDA8 ozone in 2014. The black rectangle outlines North China (NC); the red rectangle outlines South China (SC).

Figure R2. The Figure 3 in Zhao and Wang (2017), i.e., the EOF1 results in 2014 (also the sites distribution)

Since the severe air pollution events in 2013, the air pollution issues gained more attentions from the Chinese government and society, which aided to start the extensive constructions of operational monitoring stations of atmospheric components and resulted in continuous increasing number of sites (Figure S1). The number of sites in eastern China (110°E–125°E, 22°N–42°N) was 677, 937, 937, 995 and 1007 from 2014 to 2018. It is obvious that the data in 2014 were deficient, while the observations were broadly distributed in eastern China and continuously achieved since 2015. Thus, the summer O3 data from 2015 to 2018 were processed (e.g., unifying the sites and eliminating the missing value) and 868 sites in eastern China were employed here to reveal some new features of surface ozone pollutions and associated anomalous atmospheric circulations.

Although the number of sites in 2014 in our denser data source were nearly 4 times that in Zhao and Wang (2017), the data in the green box in Figure S1 were almost a blank. That is why our study period was 2015–2018. To make this point clear, we added Figure S1. From 2015 to 2018, the selected 868 sties relatively even. Certainly,

the sites were almost located around the urban area, due to their observed purposes (related to air pollutions).

---

## Author Comment (AC2) · 8 Sep 2019

**Response to Reviewer #2**

**Summary**

This manuscript aims to analyze the dominant patterns of summer ozone over China in recent years, and associated circulations. While the topic is of importance to the field, the conclusions drawn from the study are not convincing to me.

**1.** The general patterns of ozone pollution and the association with meteorology have been reported in previous papers.

**The authors need to clarify the novelty and scientific contribution of this study. The methods in general lack novelty.**

**Reply:**

The novelty of this study was sufficiently explained from three perspectives, listing as (1)-(3). Also, we revised the manuscript to present the novelty in a clearer way.

(1) In most of previous studies related to ozone pollution, the most popular topic was the relationship meteorological elements (e.g., temperature, precipitation, etc.) and  $O_3$  concentrations in single city. These kind of studies **did not included the analysis of large-scale atmospheric circulations**, the diagnosis of **dominant patterns** and their **varying features**, and the **signals for interannual variability**.

Furthermore, in the Sect. 1 (Introduction), we referred a review article published in 2017 and point out "Wang et al. (2017) reviewed the meteorological influences on ozone events, **but the referenced findings were published mainly before 2010**, when measurements in China were still scarce."

These kind of studies was quite different from our submitted manuscript. The detailed novelty was illustrated in the following point (2).

(2) Actually, Zhao and Wang (2017) also talked about the dominant pattern of surface ozone and the impact of WPSH. Here, we emphatically explained the novelty of this study by the differences with Zhao and Wang (2017). That is, using the following four sub-points, we emphasized this novelty.

(2.1) We revealed **two dominant patterns, their varying sorts** in different years and their **associated anomalous atmospheric circulations.** Although the north-south differential pattern was the first mode in 2014 (Zhao and Wang 2017), 2015 and 2016, it was sorted in the second place in 2017 and 2018 (Figure S5 in the revised manuscript). That is, our study not only revealed the two dominant patterns, what is more important, also **showed the varying features of the dominant patterns.** In the recent two years, the most dominant pattern was different from that in previous years, which is **a new feature and might related to the climate status**. Additionally, the comprehensive atmospheric circulations were analyzed, including the **location** of west Pacific subtropical high (WPSH), the **East Asia deep trough** and other atmospheric anomalies. In Wang and Zhao (2017), they solely focused on the impacts of the WPSH, particularly on the **accumulative enhancement** of WPSH.

(2.2) We clearly explained the anomalous atmospheric circulations related the  $O_3$  pollution **both in North China and in South China**. However, in Wang and Zhao (2017), the **physical mechanisms to impact O3 in North China was still not sufficiently explained** (referring to the *weak correlation* coefficients in the *green boxes* in Figure R1 d–f). We speculated the reason for insufficient explanations on  $O_3$  conditions in North China might be that the impacts form the mid-high latitudes were significant which was not involved in Zhao and Wang (2017). In our study, we found **both of the WPSH and the East Asia deep trough** had impacts on the  $O_3$  concentrations in North China where the surface  $O_3$  polluted levels were much higher than in the Yangtze River Delta and Pearl River Delta. The **WPSH and East Asia deep trough jointly modulated** the local meteorological conditions to influence the  $O_3$  concentrations.

Fig. 6. (Upper panel) the summer mean fields of meteorological parameters and (lower panel) their correlations with the daily WPSH-Lanomalies: total cloud cover (a, d), downward UV radiation at the surface (b, e), and near-surface air temperature (c, f).

**Figure R1.** The Figure 6 in Zhao and Wang (2017). The summer mean fields of meteorological parameters (a–c) and their correlations with daily WPSH: total cloud cover (a, d), UV radiation (b, e) and near surface air temperature (c, f). The added green boxes indicate the location of North China.

**Table 1.** Correlation coefficients between the time series of PAT1 (PAT2) and the key indices of atmospheric circulations and meteorological conditions. "\*\*" and "\*" indicate that the correlation coefficients were above the 99% and 95% confidence level, respectively.

| PAT1 | MDA81             | EAT 1 | $WPSH_1$          |
|------|-------------------|------------------|-------------------|
|      | $0.97^{**}$       | 0.28**           | 0.39**            |
| PAT2 | MDA8 2 | EAT 2 | WPSH 2 |
|      | 0 77**            | 0.30**           | 0.32**            |

MDA81 is the NCH-area averaged MDA8, while the MDA82 is the MDA8 difference between NC and YRD. EAT1 and EAT2 indicate the intensity of the East Asia deep trough and were calculated as the mean -Z850, shown in the black boxes in Figure 7 and Figure 9, respectively. WPSH1 ( $Z500_{(125^0E, 20^0N)} - Z500_{(125^0E, 30^0N)}$ ) and WPSH2 ( $Z500_{(110^0E, 20^0N)} - Z500_{(110^0E, 30^0N)}$ ) represents the location of WPSH.

(2.3) The number and distribution of the sites are more sufficient and updated. In the EOF analysis of Zhao and Wang (2017), the number of  $O_3$  sties was only 191 in 2014 even in a larger study region than ours (Figure R2). The number (fewer than 200) and distributions (uneven) of the sites were limited, due to the establishment progress of the observation sites of atmospheric components in 2014.

Fig. 3. The EOF1 of daily summer MDA8 ozone in 2014. The black rectangle outlines North China (NC); the red rectangle outlines South China (SC).

Figure R2. The Figure 3 in Zhao and Wang (2017), i.e., the EOF1 results in 2014 (also the sites distribution)

Since the severe air pollution events in 2013, the air pollution issues gained more attentions from the Chinese government and society, which aided to start the extensive constructions of operational monitoring stations of atmospheric components and resulted in continuous increasing number of sites (Figure S1). The number of sites in eastern China (110°E–125°E, 22°N–42°N) was 677, 937, 937, 995 and 1007 from 2014 to 2018. It is obvious that the data in 2014 were deficient, while the observations were broadly distributed in eastern China and continuously achieved since 2015. Thus, the summer O3 data from 2015 to 2018 were processed (e.g., unifying the sites and eliminating the missing value) and 868 sites in eastern China were employed here to reveal some new features of surface ozone pollutions and associated anomalous atmospheric circulations.

Although the number of sites in 2014 in our denser data source were nearly 4 times that in Zhao and Wang (2017), **the data in the green box in Figure S1 were almost a blank.** That is why our study period was 2015–2018. To make this point clear, we **added Figure S1**. From 2015 to 2018, the selected 868 sties relatively even. Certainly, the sites were almost located around the urban area, due to their observed purposes (related to air pollutions).

---

## Referee Report (RR1)

Review of "Dominant Patterns of Summer Ozone Pollution in Eastern China and Associated Atmospheric Circulations"

The authors have greatly improved the manuscript responding to the two reviewers' comments. I believe this paper is suitable for publication after minor revisions. My largest minor points are:

**Line 28-30:** Include the boxed regions shown on Figure 3a,c on the other figures of Eastern China ozone (e.g., Figure 1, 4,7,8) to assist the reader throughout the manuscript. Then here when you list the location of these regions the authors can reference Figure 1b.

**Line 111:** Do you think the high levels of MDA8 around the large cities has to do with aged pollutant transport in the cities, which may be related to lower O3 in the cities due to NOx titration but outside of the cities the air was in a different NOx regime and O3 increased? This idea has been largely absent from this paper, but I believe it is worth discussing.

**Line 116:** Severe O3 pollution is mentioned a few times in the paper but "severe" is not defined as a threshold. What do the authors mean by this? Is it anything above "Moderately polluted"?

**Line 139**: The authors focus the discussion of PAT2 on NC and YRD, with the occasional mention to PRD (e.g., line 207-208). Often there are places where the comment applies to both YRD and PRD (e.g., Lines 146, 181-202) and could be included. Is there a reason why not to include it in the discussion of PAT2?

**Lines 223-251:** I am not convinced by the discussion of Figures 10 and 11 as this is one year being compared to a really small sample size of a four-year average which includes that same one year. It is the nature of the availability of the data that the authors choose not to extend their analysis further back. I think the paper can stand alone without these two additional figures and discussion. This could be revisited after more time has passed in a later publication.

**Minor and technical comments:**

Line 22: Start off the first sentence with something like "High levels of ozone occur both in the stratosphere and at the ground level". Otherwise ozone occurs throughout the troposphere, just not always at unhealthy concentrations.

Line 34: I do not understand the significance of the greater ozone trend on the highest mountain in NC. This indicates to me that the background ozone in the free troposphere in the region is possibly increasing. Is this what the authors mean? Please clarify the significance of this statement.

Line 44:  I believe the Li et al. paper (note, add the period after 'al' in this sentence.  It is missing.  Check all references for this) uses the GEOS-Chem CTM, not the GEOS CTM.  These are different models.

Line 57:  Can the authors provide a summary sentence at the end of this paragraph linking all these studies together?

Line 58-59:  The authors claim that Wang et al. (2017) claim the study uses ozone data from prior to 2010.  How can the reader then assume that the 7 referenced studies in the paragraph above with publication dates prior to 2019 are using recent enough ozone data to that the authors can use these papers to make their claims?

Line 62:  I know I asked how your study is different to the Zhao and Wang (2017) paper, but stating "Actually, in our study, we found the …." comes out of nowhere compared to the rest of your introduction.  Please modify this sentence to be less aggressive and more like, "In this study, we built upon the previous literature analyzing ozone and meteorological influences thanks to the availability of more ozone observations by the chinese government since 2015, providing us more information to analyze then available in these earlier studies, e.g., Ziao and Wang (2017)."

Line 73: What threshold was used to unify the sites?  For example, did some sites move location but still considered as one time series?

Line 80:  Thank you for adding this detail.  However, I am not used to this type of notation; is there a reason some left brackets are curved ( and not [ ?

Line 82:  Can you provide a link either here in the text or at the end of the paper in a "Data Availability" section to where you got the ERA-Interim data.

Line 83: Remove at to read "temperature from surface to 100 hPa"

Line 86,89-90:  I am still confused by this description of reanalysis timesteps to Beijing time.  How many 6-hourly reanalysis timestamps are used in the daytime data analysis?  Why are there different time period for the 3-hourly data than 6 hourly data?  Can you use the same time period but just more 3-hourly data?  00 am to 00pm UTC does not make sense.  I think you mean 00 UTC to 12 UTC which would mean you have 3 6-hourly timesteps (00, 06, 12 UTC). The am and pm should also be removed from the 21 UTC to 09 UTC.  And this means you have 5 3-hourly timesteps (21, 00, 03, 06, 09 UTC).  Why is this offset 3 hours from the 6-hourly timesteps (could have done 00,03,06,09, 12 UTC)?

Line 94:  remove 'ly' from relatively

Line 104: I suggest changing this sentence to "….was mostly lower than 100 ug/m3, and lower than the O3 pollution in North China and in the Huanghuai area (NCH, about 31-41N, 110-

120E)". This modifies the second half of the sentence as well as adding in the box region for the NCH (which I guessed from Figure 3).

Line 112: are cities large and small based on population or area-covered?

Line 115: Could highlight that this threshold is matching the list now given in line 80-81; e.g., change to ….the threshold of "heavily O3 pollution" in China…

Line 118: Are these cities with large populations "Megacities"?

Line 119: How often was the MDA8 value nearly above 100 ug/m3? Are you trying to say the ozone was hardly below 100 ug/m3 on any day in the time series?

Line 122: What does "exceeded the health threshold" in reference to? Is that "Good" and above?

Line 127: Change reference to (Figure 2a,b)

Line 141: There looks to be extra spacing in front of -1x

Line 142: Change showed to shows. There are other places where the authors switch back and forth between verb tenses (e.g., Line 154, 159).

Line 145: add (Fig 4a,b) after "respectively".

Line 149: change recent four years to "the four years of study" since this becomes less true after publication.

Line 150: What does "were reasonably supposed to be relatively stable on the daily time-scale" mean. I think change to "Despite the economic productions and human activities steadily increasing from 2015 to 2018 in Eastern China, we assume the emissions of ozone precursors to be relatively stable on the daily time-scale".

Line 154-156: I do not feel this "For example, " sentence is necessary.

Line 158: so this is stagnation and as such there is the buildup of pollutants too without removal from weather systems passing by. I do not think that idea has been discussed.

Line 159: Start sentence with "In Figure 5a, there are negative …"

Line 163: Change to "(EAT, Table 1)"

Line 166: This final part might be a bit of a stretch.

Line 184:  In the author's comments to my first review, they explain to me why they choose different pressure levels for Figure 5a and 6a, but please include the reasoning in the paper.

Line 186:  It is really difficult to find Chukchi Peninsula on the map given the country lines are in grey.  Can you include lat/lon of the region you mean because I think the peninsula is mostly under negative Z500.

Line 188: Similarly, I would argue that it was limited to the east of Korea so again add coordinates to ensure your reader is looking where you want them to be looking.

Line 194:  Remove the word obvious.

Line 204: Change "the past four years" to "these past four years" as this will change after publication.

Line 217: Could include a reference to Fig 8.

The following are the reasons why I do not think Figures 10 and 11 add to the manuscript:

Line 223: change to "were generally negative".  I think there is a mix of positive and negative in the YRD in Figure 7b, not strictly positive as is written.

Line 227: Again, I see a ridge over Chukchi Peninsula, not a trough, so include coordinates of where you want the reader to be looking.

Line 229: Both Figure 10 and 11 could have the same purple arrows to identify how the authors would calculate the WPSH as in Figures 5 and 6.  Again, authors could include coordinates or a box region as in Figures 5 and 6 to indicate the location of EAT and WPSH.

Line 231: I disagree with the authors that more low to mid-level clouds formed as it looks to me that the box straddles the zero line and that instead of a decrease in solar radiation reaching the ground in NC I see positive colors in that box too in Figure 10d.

Line 236: What do the authors mean by "south of YRD"?  Do they mean outside of the box or the southern portion of the YRD?  Does that mean PRD?

Lines 225-237:  There are no significant differences near the NC or YRD in any of the four panels in Figure 10.

Line 238: I do not like how the sentence starts with "-+-".  Can the authors at least put "The" prior to the symbols?  Is this the same as an OMEGA block and could be written as such instead of -+- ?

Line 243:  I do not see an anomalous anticyclone over the NCH in Figure 11a, but more on the border with cyclonic flow to the southwest and anticyclone to the east, and as the authors describe later in reference to the water vapor flux, the NCH is in a region of anomalous divergent flow.

Line 244:  When the authors say "it was difficult to form cloud" the NCH straddles the zero line and shows a mix of both positive and negative SSR so how can they say "more solar radiation directly reach the ground (Figure 11d)"?

Line 247:  Out of nowhere the authors mention "signals of global warming".  Such bold statements in a manuscript, which depend on figures in the supplemental material, should not be made.

Overall the Discussion is fine.
Line 254: Add a reference (e.g. Li et al., 2018) to the end of the opening sentence, possibly bringing over the point made at the end of the discussion on line 281 "At present, the fine PM decreased in the summers in eastern China….".

Line 264: the increased SAT and thus decreased cold air advection from the north a) seems backwards (less cold air advection would likely lead to increased SAT at lower latitudes) and b) the negative Z500 to the west with flow from south and southwest likely brought warmer temperatures from the southern latitudes into the region.

Line 270:  "…daily emission data were difficult to be acquired" implies the authors have acquired such data.  Is that true or are these data difficult to acquire?

Line 273: remove the comma before (2019).

Line 275:  How were the "domestic anthropogenic emissions alone would have led to ozone decreases" determined by the Lu et al. study?

Line 277:  Is "were still unclear" in reference to your work presented here or in reference to the Lu et al. study?

Figure comments:

Figure 2:  Can NC, NCH, YRD, and PRD be added to the Figure caption in a similar fashion to how it was added to the Comments to the Reviewer (Page 47).  Also, could add a reference to Figure S2 in the caption to remind readers to look there for the locations of these cities.

Figure 3:  Add to Figure 3 caption what the dashed lines in panels b and d represent.  I assume horizontal are the standard deviation and the vertical separate the years.

Figure 5, 6, 10, 11:  The units for water vapor flux are written as kg*m/(kg*s) which is a bit odd. I strongly encourage the authors to use negative values to indicate denominators in units throughout the paper.  This would change these units to kg m kg$^{-1}$ s$^{-1}$ which then leads me to my next question whether these are the correct units.  In its current form are the kg's representing different things (dry air vs moisture?)

Figure 7:  It is difficult to see the crosses when the O3 anomaly is less than -25 ppb (the dark blue color).  Can that be adjusted? Or maybe have white crosses instead?  Also the y-axis minor tick marks are an odd spacing compared to the major 5 degree labelled tick marks.  Can these minor tick marks be changed to maybe every 1 degree?

Figure 9:  Are the variations in reference to an average for all available sites in the NC and the YRD regions?  If so, please add this level of detail to the Figure 9 caption.

---

## Author Response (AR2)

**Response to Reviewer #1**

The authors have greatly improved the manuscript responding to the two reviewers' comments. I believe this paper is suitable for publication after minor revisions.

*Largest minor points:*

**Line 28-30: Include the boxed regions shown on Figure 3a,c on the other figures of Eastern China ozone (e.g., Figure 1, 4, 7, 8) to assist the reader throughout the manuscript. Then here when you list the location of these regions the authors can reference Figure 1b.**

*Reply:*

According to the features of each Figure, the mentioned boxes were also included in Figure 1, 4, 7 and 8.

When we listed the locations, e.g., NC, YRD and PRD, Figure 1b was referenced.

*Revision:*

......the high O₃ concentrations in China are mainly observed in urban regions, such as in North China (NC, Figure 1b), the Yangtze River Delta (YRD) and the Pearl River Delta (PRD) where rapid......

[Figure]

**Figure 1.** Distribution of the (a) mean values and (b) maximum values of MDA8 (Unit: μg/m³) at the observation sites in summer from 2015 to 2018. The black boxes in panels a and b indicated the locations of North China and Huanghuai region (NCH), North China (NC), Yangtze River Delta (YRD) and Pearl River Delta (PRD).

[Figure]

**Figure 2.** Anomalies of the summer mean MDA8 (Unit: μg/m³) in 2015 (a), 2016 (b), 2017 (c) and 2018 (d), relative to the mean during 2015–2018. The black pluses indicate that the maximum MDA8 was larger than 265 μg/m³. The black boxes in panel b indicated the locations of NC, YRD and PRD, while that in panel d was the NCH area.

[Figure]

**Figure 5.** Composites of the MDA8 (Unit: μg/m³) for PAT1 (a, b) and PAT2 (c, d) in summer from 2015 to 2018. Panels (a) and (c) were composited when the time coefficient of EOF1 and EOF2 was greater than one standard deviation, while panels (b) and (d) were composited when the time coefficient was less than –1×one standard deviation. The black box in panel a-b indicated the location of NCH, while those in panel c-d were the NH, YRD and PRD area.

[Figure]

**Figure 8**. The first (a, c, e, g) and second (b, d, f, h) EOF spatial patterns of MDA8 in summer in 2015 (a, b), 2016 (c, d), 2017, (e, f) and 2018(g, h). The percentage number in panels (a, c, e, g) and (b, d, f, h) are the variance contributions of the first and second EOF mode. The black boxes indicated the location of NCH, NH, YRD and PRD, respectively.

**Line 111: Do you think the high levels of MDA8 around the large cities has to do with aged pollutant transport in the cities, which may be related to lower O3 in the cities due to NOx titration but outside of the cities the air was in a different NOx regime and O3 increased? This idea has been largely absent from this paper, but I believe it is worth discussing.**

*Reply:*

The related discussions were added in the revised manuscripts, e.g., in the Introduction, the first paragraph of Section 3, and in the Conclusions and Discussions.

*Revision in the Introduction:*

……Although deep stratospheric intrusions may elevate surface ozone levels (Lin et al., 2015), the main source of surface ozone is the photochemical reactions between the oxides of nitrogen ($NO_x$) and volatile organic compounds (VOC), i.e., $NO_x$ + VOC = $O_3$. The concentrations of $NO_x$ and VOC are fundamental drivers impacting ozone production, and are sensitive to the regime of ozone formation, i.e., $NO_x$-limited or VOC-limited (Jin and Holloway 2015)……

*Revision in the first paragraph of Section 3:*

……Surface $O_3$ pollution was closely linked to the anthropogenic emissions that dispersed and concentrated in the large cities (Fu et al., 2012), which was similar to the haze pollution (Yin et al., 2015). In the megacity cluster, the photochemical regime for ozone formation is combination of $NO_x$-limited and VOC-limited regimes (Jin and Holloway 2015). In the YRD and PRD, high levels of MDA8 were scattered around the large cities. Due to high emissions of $NO_x$ both in large and small cities in the NCH region, the high-level $O_3$ values were contiguous, indicating extensively surface $O_3$ pollution (Figure 1)……

*Revision in the Conclusions and Discussions:*

……In this study, we mainly emphasized the contribution of the meteorological impacts and assumed the emissions of ozone precursors were relatively stable on the daily time-scale. Observational and modelling studies suggested that photochemical production of ozone in the NC, YRD and PRD was the transitional regime (i.e., both

reductions of $NO_x$ and VOC would reduce $O_3$), which would influence the concentrations of surface ozone (Jin and Holloway 2015). There is no doubt that the human activities were the fundamental driver of air pollution even on the daily time-scale, thus the joint effects of the daily meteorological conditions and anthropogenic emissions (including the photochemical regimes) needed to be discussed in future work……

**Line 116: Severe O3 pollution is mentioned a few times in the paper but "severe" is not defined as a threshold. What do the authors mean by this? Is it anything above "Moderately polluted"?**

*Reply:*

Most of the uses of "severe" were replaced by more accurate presentations, such as "high levels of $O_3$ pollutions", "heavily polluted $O_3$ pollutions" and "high concentrations of $O_3$", etc.

*Revision :*

Shanghai in YRD, Guangdong and Zhongshan in PRD (Figure S2). These cities had large populations and were with  high levels of $O_3$ pollutions. In Beijing, Tianjin and Tangshan, the MDA8 values were nearly above 100 µg/m³ and frequently

distribution of the mean MDA8, the maximum MDA8 to the south of 30°N was lower by comparison. Although approximately 60% of summer days were non-$O_3$-polluted in the cities of Guangzhou and Zhongshan (Figure 2d), heavily polluted  $O_3$ pollution also occurred in the PRD (Figure 1b). ↵

and <120 µg/m³ for PAT1P and PAT1N, respectively. For the second pattern, the PAT2P appeared as a diminishing pattern from the north to the south (Figure 4c), however, there was  high concentrations of ozone pollution in the YRD and PRD under PAT2N conditions (Figure 4d). Therefore, the centres of $O_3$ variation were NCH for the PAT1, and NC and the YRD for

**Line 139: The authors focus the discussion of PAT2 on NC and YRD, with the occasional mention to PRD (e.g., line 207-208). Often there are places where the comment applies to both YRD and PRD (e.g., Lines 146, 181-202) and could be included. Is there a reason why not to include it in the discussion of PAT2?**

*Reply:*

From the mean MDA8 (Figure 1a), the composites for PAT2 (Figure 4), the ozone concentrations in PRD were not as high as those in the NC and YRD areas. Thus, we did not pay much attentions on PRD region. Particularly, in the "Associated atmospheric circulations", we did not mention the ozone pollution in PRD.

However, when analyzing the observational features, we also included the ozone in PRD to show some new features.

**Lines 223-251: I am not convinced by the discussion of Figures 10 and 11 as this is one year being compared to a really small sample size of a four-year average which includes that same one year. It is the nature of the availability of the data that the authors choose not to extend their analysis further back. I think the paper can stand alone without these two additional figures and discussion. This could be revisited after more time has passed in a later publication.**

*Reply:*

According to the reviewer's comment, we decided to cancel the Figure 10 and 11. We now focus on the dominant patterns and their varying features in different years. Some new works will be supplemented and we will revisit the Figure 10 and Figure 11 in a later manuscript.

The contents related to Figure 7–9 were rewritten and redistributed in the revised version. The texts associated Figure 10 &11 were deleted. Detailed revisions can be found in the revised manuscript and also the mark-up manuscript.

*Minor and technical comments:*

**Line 22: Start off the first sentence with something like "High levels of ozone occur both in the stratosphere and at the ground level". Otherwise ozone occurs throughout the troposphere, just not always at unhealthy concentrations.**

*Reply:*

According to the reviewer's comment, this sentence was revised.

*Revision:*

High levels of zone occurs both in the stratosphere and at the ground level. Stratospheric ozone forms a protective layer at shields us from the sun's harmful ultraviolet radiation. However, surface ozone is an air pollutant and has harmful effects

**Line 34: I do not understand the significance of the greater ozone trend on the highest mountain in NC. This indicates to me that the background ozone in the free troposphere in the region is possibly increasing. Is this what the authors mean? Please clarify the significance of this statement.**

*Reply:*

What we meant is that the increasing trend was widespread in the east of China and it is meaningful to study the variations in ozone pollution in this region. Furthermore, after your reminder, we thought the increasing trend partly indicate the background ozone in the free troposphere in the region is possibly increasing.

Referencing the reviewer's suggestion, we revised the statement as follows.

*Revision:*

……Although far away from the anthropogenic emissions, the summer (June-July-August, JJA) $O_3$ on the highest mountain over NC (Mount Tai) increased significantly by 2.1 ppbv $yr^{-1}$ from 2003 to 2015 (Sun et al., 2016)……

**Line 44: I believe the Li et al. paper (note, add the period after 'al' in this sentence. It is missing. Check all references for this) uses the GEOS-Chem CTM, not the GEOS CTM.   These are different models.**

*Reply:*

We referenced the name of the model used by Li et al. (2018) in their Abstract.

tions thought to prevail in urban China while decreasing ozone under rural $NO_x$-limited conditions. However, simulations with the Goddard Earth Observing System Chemical Transport Model (GEOS-Chem) indicate that a more important factor for ozone trends in the North China Plain is the ~40% decrease of fine particulate matter ($PM_{2.5}$) over the 2013–2017 period, slowing down the aerosol sink of hydroperoxy ($HO_2$) radicals and thus stimulating ozone production.

After carefully checking their publication, we found the detailed name of the model in the Mechods. The sentence was revised as follows:

*Revision:*

……Employed the GEOS-Chem chemical transport model, Li et al. (2018) found that rapid decreases in fine particulate matter levels significantly stimulated ozone production in NC by slowing down the aerosol sink of hydro-peroxy radicals……

**Line 57: Can the authors provide a summary sentence at the end of this paragraph linking all these studies together?**

*Reply:*

According to the reviewer's comment, a summary sentence was supplemented.

*Revision:*

……Thus, in addition to human activities and secondary aerosol processes, the impacts of atmospheric circulations and meteorological conditions must be systematically studied to improve understanding of the O$_3$ pollution in North China……

**Line 58-59: The authors claim that Wang et al. (2017) claim the study uses ozone data from prior to 2010. How can the reader then assume that the 7 referenced studies in the paragraph above with publication dates prior to 2019 are using recent enough ozone data to that the authors can use these papers to make their claims?**

*Reply:*

To avoid the confusions, we deleted this sentence.

*Revision:*

 Since 2015, O$_3$ measurements in eastern China were

**Line 62: I know I asked how your study is different to the Zhao and Wang (2017) paper, but stating "Actually, in our study, we found the …." comes out of nowhere compared to the rest of your introduction. Please modify this sentence to be less aggressive and more like, "In this study, we built upon the previous literature**

**analyzing ozone and meteorological influences thanks to the availability of more ozone observations by the Chinese government since 2015, providing us more information to analyze then available in these earlier studies, e.g., Zhao and Wang (2017)."**

*Reply:*

Thanks to the kind remind from the reviewer. We worried too much to show the differences between our studies and Zhao and Wang (2017) and now follow the reviewer's suggestions.

*Revision:*

The dominant patterns of daily ozone in summer in east of China are still unclear. In this study, we built upon the previous literature analysing ozone and meteorological influences thanks to the availability of more ozone observations by the Chinese government since 2015, providing us more information to analyse than available in these earlier studies, e.g., Zhao and Wang (2017).

**Line 73: What threshold was used to unify the sites? For example, did some sites move location but still considered as one time series?**

*Reply:*

The sites with missing data >5% were removed and there were 868 sited were kept in the four years.

**Line 80: Thank you for adding this detail. However, I am not used to this type of notation; is there a reason some left brackets are curved ( and not [ ?**

*Reply:*

It is accurate mathematical notations. That is, [ contained the boundary value, however, ( did not contained it.

[Figure]

**Line 82: Can you provide a link either here in the text or at the end of the paper in a "Data Availability" section to where you got the ERA-Interim data.**

*Reply:*

Data Availability was added as the ACP journal required.

*Revision:*

*Data availability.*

Hourly $O_3$ concentration data is supported by the website: http://beijingair.sinaapp.com (Ministry of Environmental Protection of China, 2018). Atmospheric circulation datasets are downloaded from http://www.ecmwf.int/en/research/climate-reanalysis/era-interim (ERA-Interim, 2018).

**Line 83: Remove at to read "temperature from surface to 100 hPa".**

*Reply:*

The errors were corrected.

*Revision:*

air temperature  from surface to 100 hPa,

**Line 86, 89-90: (1) I am still confused by this description of reanalysis timesteps to Beijing time. How many 6-hourly reanalysis timestamps are used in the daytime data analysis?   Why are there different time period for the 3-hourly data than 6 hourly data? Can you use the same time period but just more 3-hourly data?**
**(2) 00 am to 00pm UTC does not make sense. I think you mean 00 UTC to 12 UTC which would mean you have 3 6-hourly timesteps (00, 06, 12 UTC). The am and pm should also be removed from the 21 UTC to 09 UTC. And this means you have 5 3hourly timesteps (21, 00, 03, 06, 09 UTC). Why is this offset 3 hours from the 6-hourly timesteps (could have done 00,03,06,09, 12 UTC)?**

*Reply:*

(1) The required answers were clearly shown on the website of ERA-Interim (https://confluence.ecmwf.int/pages/viewpage.action?pageId=56658233). On the page

about "ERA-Interim: 'time' and 'steps', and instantaneous, accumulated and min/max parameters", we can found the explanations about the parameters.

ERA-Interim data is archived at website differently according to whether they are produced by the **analysis (An), or the forecast (Fc)**, and timesteps of data which we could get is shown in Fig R1. Only precipitation and surface solar are produced by forecast in all the data which we used, which was illustrated in Table 8 and 9 of Berrisford et al. (2011). Fc data is the **accumulated** (from the beginning of the forecast) and can be treated as 3-hr data due to 4 timesteps in 12 hours, and An data is **instantaneous** (Berrisford et al., 2011).

We wanted to research the link between atmospheric circulation and MDA8 which mostly occurs in daytime (8:00 a.m. to 8:00 p.m. Beijing Time). For Fc, we considered **00 UTC to 12UTC (8:00 to 20:00 Beijing Time)**, the upper way in Figure R1, (first '+12' line in Fig. R1) as daytime in Beijing. It means we just use 1 timesteps data which is accuunulated from 00 UTC to 12 UTC. But for An (second line in Fig. R1), we could only calculate the mean of 00:00 and 06:00 UTC as daytime mean and it represent mean of the time-scale 3-hour (half of the timesteps) before the 00:00 to 3-hour after 06:00. It means that daytime is **21 UTC to 9 UTC (5:00 to 17 Beijing Time).**

[Figure]

Figure R1. ERA-interim Data timesteps (downloaded from the ERA website).

Reference:
Berrisford, P, Dee, DP, Poli, P, Brugge, R, Fielding, M, Fuentes, M, Kållberg, PW, Kobayashi, S, Uppala, S, Simmons, A, The ERA-Interim archive Version 2.0. https://www.ecmwf.int/node/8174.

(2) The timing way was changed to 24HR way.

*Revision:*

……Due to the different representative period of each element in ERA-Interim data, the daytime for Z, wind, relative humidity, vertical velocity, air temperature and cloud cover was from 05 to 17 (Beijing Time; 21–09 UTC), while it is from 08 to 20 (Beijing Time; 00 to 12 UTC) for precipitation and downward solar radiation……

**Line 94: remove 'ly' from relatively**

*Reply:*

The errors were corrected.

*Revision:*

extracted the relative change features of the original data on the daily time-scale.

**Line 104: I suggest changing this sentence to "….was mostly lower than 100 ug/m3, and lower than the O3 pollution in North China and in the Huanghuai area (NCH, about 31-41N, 110-120E)". This modifies the second half of the sentence as well as adding in the box region for the NCH (which I guessed from Figure 3).**

*Reply:*

The errors were corrected.

*Revision:*

South China), the mean MDA8 was mostly lower than 100 μg/m³ and  lower than  the $O_3$ pollution in North China and in the Huanghuai area (NCH, Figure 1a). It is notable that, although the values of MDA8 in the

**Line 112: are cities large and small based on population or area-covered?**

*Reply:*

The large and small cites are distinguished based on population.

**Line 115: Could highlight that this threshold is matching the list now given in line 80-81; e.g., change to ….the threshold of "heavily O3 pollution" in China…**

*Reply:*

According to the reviewer's advice, this sentence was improved.

*Revision:*

Furthermore, the maximum values of MDA8 for four summers were extracted to evaluate the severest levels of $O_3$ pollution (Figure 1b). To the north of 30ºN, the maximum MDA8 at most sites was above 265µg/m³ (i.e., the threshold of heavily $O_3$ pollution in China), indicating that  heavily $O_3$ pollution had occurred.

 The observed summer MDA8 anomalies in

**Line 118: Are these cities with large populations "Megacities"?**

*Reply:*

Yes, these cities with large populations are megacities. To emphasize the large populations, we maintained the presentation.

**Line 119: How often was the MDA8 value nearly above 100 ug/m3? Are you trying to say the ozone was hardly below 100 ug/m3 on any day in the time series?**

*Reply:*

The detailed percentages were explained in the next sentence.

The percentage of non-$O_3$-polluted days (<100 µg/m³) and moderate $O_3$-polluted days (>215 µg/m³) were 14.9% and 15.5% for the mean MDA8 of these three cities.

**Line 122: What does "exceeded the health threshold" in reference to? Is that "Good" and above?**

*Reply:*

It is the upper limit "Excellent" level.

*Revision:*

µg/m³) were 14.9% and 15.5% for the mean MDA8 of these three cities. The former percentage indicated that more than 85%

$O_3$ concentrations exceeded the health threshold (i.e., the upper limit "Excellent" level ), and the later meant, in more than 15%

**Line 127: Change reference to (Figure 2a,b)**

*Reply:*

The sentence was corrected.

*Revision:*

to those of Beijing and Tianjin in 2017 and 2018 (Figure 3a, b). In Nanjing and Shanghai, the MDA8 did not show a clear

increasing trend (Figure 3c). Similar to the distribution of the mean MDA8, the maximum MDA8 to the south of 30°N was

**Line 141: There looks to be extra spacing in front of -1x**

*Reply:*

The errors were corrected.

*Revision:*

……less than $-1 \times$one standard deviation…..

**Line 142: Change showed to shows. There are other places where the authors switch back and forth between verb tenses (e.g., Line 154, 159).**

*Reply:*

Similar errors were checked and corrected throughout the manuscript.

**Line 145: add (Fig 4a, b) after "respectively".**

*Reply:*

The reference of Figure were added.

*Revision:*

and <120 µg/m³ for PAT1P and PAT1N, respectively (Figure 5a, b). For the second pattern, the PAT2P appeared as a

diminishing pattern from the north to the south (Figure 5c), however, there was  high concentrations of ozone pollution

**Line 149: change recent four years to "the four years of study" since this becomes less true after publication.**

*Reply:*

The errors were corrected.

*Revision:*

In eastern China, the economic productions and human actives steadily developed in the  four years of study and the

emissions of ozone precursors were reasonably supposed to be relatively stable on the daily time-scale. Differently, the daily

**Line 150: What does "were reasonably supposed to be relatively stable on the daily time-scale" mean. I think change to "Despite the economic productions and human activities steadily increasing from 2015 to 2018 in Eastern China, we assume the emissions of ozone precursors to be relatively stable on the daily time-scale".**

*Reply:*

According to the reviewer's advice, this sentence was improved.

*Revision:*

In eastern China, despite the economic productions and human actives steadily increased in the four years of study and we assume the emissions of ozone precursors to be relatively stable on the daily time-scale.

**Line 154-156: I do not feel this "For example," sentence is necessary.**

*Reply:*

This sentence was deleted.

*Revision:*

(PAT2P composite minus PAT2N composite)  are shown Figure  7.  For the first

**Line 158: so this is stagnation and as such there is the buildup of pollutants too without removal from weather systems passing by. I do not think that idea has been discussed.**

*Reply:*

The removal of ozone pollutants by the weather were clearly discussed in the revised manuscript, which make our study more complete.

*Revision in the Abstract:*

[revised manuscript text omitted]

**Line 184: In the author's comments to my first review, they explain to me why they choose different pressure levels for Figure 5a and 6a, but please include the reasoning in the paper.**

*Reply:*

According to the reviewer's advice, related contents were included.

*Revision:*

between NC and the YRD was 0.77 (Table 1). The impacts of atmospheric circulations on the photochemical reactions and removal effects in the above two areas are analysed in Figure 67. It is notable that the signals of atmospheric circulations were clearer at the lower troposphere (i.e., 850 hPa) for PAT1 (Figure 6a), however, the signals for PAT2 could be recognized both at the low- and mid- troposphere (Figure 7a). Due to the broad positive Z500 anomalies at the high latitudes of Eurasia, the

**Line 186: It is really difficult to find Chukchi Peninsula on the map given the country lines are in grey. Can you include lat/lon of the region you mean because I think the peninsula is mostly under negative Z500.**

*Reply:*

The lat/lon of Chukchi Peninsula was included.

*Revision:*

……from the Chukchi Peninsula (about cantering at $180\,^{\circ}$E, $66.5\,^{\circ}$N) to Northeast China……

**Line 188: Similarly, I would argue that it was limited to the east of Korea so again add coordinates to ensure your reader is looking where you want them to be looking.**

*Reply:*

According to the reviewer's advice, this sentence was improved.

*Revision:*

……The East Asia deep trough was stronger (R=0.3), but was limited to the Sea of Japan.

**Line 194: Remove the word obvious.**

*Reply:*

According to the reviewer's advice, this sentence was improved.

*Revision:*

YRD region (Figure 7b). A moist-cool environment, weak solar radiation and  wet deposition reduced the ozone

**Line 204: Change "the past four years" to "these past four years" as this will change after publication.**

*Reply:*

According to the largest minor comment to Figure 10 & 11, the related sentence was already deleted.

**Line 217: Could include a reference to Fig 8.**

*Reply:*

According to the largest minor comment to Figure 10 & 11, the related sentence was already deleted.

**The following are the reasons why I do not think Figures 10 and 11 add to the manuscript:**

**Line 223: change to "were generally negative". I think there is a mix of positive and negative in the YRD in Figure 7b, not strictly positive as is written.**

**Line 227: Again, I see a ridge over Chukchi Peninsula, not a trough, so include coordinates of where you want the reader to be looking.**

**Line 229: Both Figure 10 and 11 could have the same purple arrows to identify how the authors would calculate the WPSH as in Figures 5 and 6. Again, authors could include coordinates or a box region as in Figures 5 and 6 to indicate the location of EAT and WPSH.**

**Line 231: I disagree with the authors that more low to mid-level clouds formed as it looks to me that the box straddles the zero line and that instead of a decrease in**

solar radiation reaching the ground in NC I see positive colors in that box too in Figure 10d.

Line 236: What do the authors mean by "south of YRD"? Do they mean outside of the box or the southern portion of the YRD? Does that mean PRD?

Lines 225-237: There are no significant differences near the NC or YRD in any of the four panels in Figure 10.

Line 238: I do not like how the sentence starts with "-+-". Can the authors at least put "The" prior to the symbols? Is this the same as an OMEGA block and could be written as such instead of -+- ?

Line 243: I do not see an anomalous anticyclone over the NCH in Figure 11a, but more on the border with cyclonic flow to the southwest and anticyclone to the east, and as the authors describe later in reference to the water vapor flux, the NCH is in a region of anomalous divergent flow.

Line 244: When the authors say "it was difficult to form cloud" the NCH straddles the zero line and shows a mix of both positive and negative SSR so how can they say "more solar radiation directly reach the ground (Figure 11d)"?

Line 247: Out of nowhere the authors mention "signals of global warming". Such bold statements in a manuscript, which depend on figures in the supplemental material, should not be made.

*Reply:*

According to the largest minor comment to Figure 10 & 11, the related sentence was already deleted.

We now focus on the dominant patterns and their varying features in different years. Some new works will be supplemented and we will revisit the Figure 10 and Figure 11 in a later manuscript. **We believe the above comments from the reviewer will be helpful in our new researches.**

The contents related to Figure 7–9 were rewritten and redistributed in the revised version. The texts associated Figure 10 &11 were deleted. Detailed revisions can be found in the revised manuscript and also the mark-up manuscript.

**Overall the Discussion is fine.**

**Line 254: Add a reference (e.g. Li et al., 2018) to the end of the opening sentence, possibly bringing over the point made at the end of the discussion on line 281 "At present, the fine PM decreased in the summers in eastern China…."**

*Reply:*

According to the reviewer's advice, this sentence was improved.

*Revision:*

At present, the fine particulate matter decreased in the summers in eastern China, and ground-level ozone pollution became the major air challenge in the summers in the east of China (Li et al., 2018).

**Line 264: the increased SAT and thus decreased cold air advection from the north a) seems backwards (less cold air advection would likely lead to increased SAT at lower latitudes) and b) the negative Z500 to the west with flow from south and southwest likely brought warmer temperatures from the southern latitudes into the region.**

*Reply:*

According to the reviewer's advice, this sentence was improved.

*Revision:*

……positive geopotential height anomalies at the high latitudes significantly decreased cold air advection from the north and thus increased the surface air temperature……

**Line 270: "…daily emission data were difficult to be acquired" implies the authors have acquired such data. Is that true or are these data difficult to acquire?**

*Reply:*

It is true these are difficult to acquire. To avoid such confusions, we revised this sentence.

*Revision:*

……There is no doubt that the human activities were the fundamental driver of air

pollution even on the daily time-scale, thus the joint effects of the daily meteorological conditions and anthropogenic emissions needed to be discussed in future work……

**Line 273: remove the comma before (2019).**

*Reply:*

The errors were corrected.

*Revision:*

……Lu et al. (2019) found that……

**Line 275: How were the "domestic anthropogenic emissions alone would have led to ozone decreases" determined by the Lu et al. study?**

*Reply:*

Lu et al. (2019) based on series of GEOS-Chem experiments and gave such conclusions.

> ozone chemical production. Our results indicate that there would be no days with MDA8 ozone > 80 ppbv in these major Chinese cities in the absence of domestic anthropogenic emissions. We find that the 2017 ozone increases relative to 2016 are largely due to higher background ozone driven by hotter and drier weather conditions, while changes in domestic anthropogenic emissions alone would have led to ozone decreases in 2017. Meteorological conditions in 2017 favor natural source contributions (particularly soil $NO_x$ and

According to the reviewer's advice, this sentence was improved.

*Revision:*

2017 surface ozone increases relative to 2016 in China are largely due to hotter and drier weather conditions, while changes in domestic anthropogenic emissions alone would have led to ozone decreases in 2017 basing on their GEOS-Chem experiments.

**Line 277: Is "were still unclear" in reference to your work presented here or in reference to the Lu et al. study?**

*Reply:*

According to the reviewer's advice, this sentence was improved.

*Revision:*

……The simultaneous large-scale atmospheric circulations on an interannual scale and their possible preceding climate drivers, e.g., sea ice, and sea surface temperature, were **still unclear so far**……

***Figure comments:***

**Figure 2: Can NC, NCH, YRD, and PRD be added to the Figure caption in a similar fashion to how it was added to the Comments to the Reviewer (Page 47). Also, could add a reference to Figure S2 in the caption to remind readers to look there for the locations of these cities.**

*Reply:*

According to the reviewer's advice, this Figure caption was improved.

*Revision:*

**Figure 3.** Variations in MDA8 (Unit: μg/m³) of polluted cities from 2015 to 2018, including (a) Beijing **(capital of China)**, Tianjin and Tangshan **near the capital city**; (b) Taiyuan, Weifang and Shijiazhuang **in the south of NCH**; (c) Shanghai and Nanjing **in YRD**; and (d) Zhongshan and Guangzhou **in PRD**. The cities in panels (a)-(d) were located from north to south **and were illustrated in Figure S2**. The horizontal dash lines indicated the value of 100 μg/m³ and 215 μg/m³.

**Figure 3: Add to Figure 3 caption what the dashed lines in panels b and d represent. I assume horizontal are the standard deviation and the vertical separate the years.**

*Reply:*

According to the reviewer's advice, this Figure caption was improved.

*Revision:*

**Figure 4.** The first EOF pattern (PAT1: a, b) and second EOF pattern (PAT2: c, d) of MDA8 in summer from 2015 to 2018, including the spatial pattern (a, c) and the time coefficient (b, d). The black boxes in panels a and c are the selected North China and Huanghuai region (NCH), North China (NC), Yangtze River Delta (YRD) and Pearl River Delta (PRD). The EOF analysis were applied to the daily MDA8 anomalies at 868 stations to extract the relatively change features of the original data on the daily time-scale. The percentages on panel (b) and (d) were the variance contributions of the first and second EOF mode. **The horizontal dash lines indicated one standard deviation, and the vertical ones separated the years.**

**Figure 5, 6, 10, 11: The units for water vapor flux are written as kg*m/(kg*s) which is a bit odd. I strongly encourage the authors to use negative values to indicate denominators in units throughout the paper. This would change these units to kg m kg-1 s-1 which then leads me to my next question whether these are the correct units. In its current form are the kg's representing different things (dry air vs**

**moisture?)**

*Reply:*

We correct units as gcm$^{-1}$s$^{-1}$hpa$^{-1}$ which is widely adopted in research.

*Revision:*

[Figure]

**Figure 6.** Differences of the daytime atmospheric circulations (i.e., PAT1P minus PAT1N). (a) Geopotential height at 850 hPa (Unit: 10gpm, contours) and surface air temperature (Unit: K, shading), (b) water vapor flux (Unit:gs$^{-1}$cm$^{-1}$hpa$^{-1}$) at 850 hPa (arrows) and precipitation (Unit: mm, shading), (c) 100°E–120°E mean wind (Unit: m/s, arrows) and relative humidity (Unit: %, shading), (d) downward solar radiation at the surface (Unit:10$^7$ J/m$^2$, shading) and the sum of low and medium cloud cover (Unit: 1, contours). The white dots indicate that the shading was above the 95% confidence level. The green boxes in panels (a), (b) and (d) show the NCH region, and the black box in panel (a) indicates the location of the East Asia trough. The purple triangles in panel (a) indicated the data used to calculate the WPSH$_1$, while the red triangle represented the west ridge point of WPSH.

[Figure]

**Figure 7.** Differences of the daytime atmospheric circulations (i.e., PAT2P minus PAT2N). (a) Geopotential height at 500 hPa (Unit: 10gpm, contours) and surface air temperature (Unit: K, shading), (b) water vapor flux (Unit:$gs^{-1}cm^{-1}hpa^{-1}$) at 850 hPa (arrows) and precipitation (Unit: mm, shading), (c) 100°E–120°E mean wind (Unit: m/s, arrows) and relative humidity (Unit: %, shading), (d) downward solar radiation at the surface (Unit: $10^7J/m^2$, shading) and the sum of low and medium cloud cover (Unit: 1, contours). The white dots indicate that the shading was above the 95% confidence level. The green boxes in panel (a), (b) and (d) are the NC and YRD regions, and the black box in panel (a) indicates the location of the East Asia trough. The purple triangles in panel (a) indicated the data used to calculate the WPSH$_2$.

**Figure 7: It is difficult to see the crosses when the O3 anomaly is less than -25 ppb (the dark blue color). Can that be adjusted? Or maybe have white crosses instead? Also the y-axis minor tick marks are an odd spacing compared to the major 5 degree labelled tick marks. Can these minor tick marks be changed to maybe every 1 degree?**

*Reply:*

According to the reviewer's advice, the Figure 5 was re-plotted. For example, the color bar was re-scaled and the label marks of y-axis was re-divided.

*Revision:*

[Figure]

**Figure 2.** Anomalies of the summer mean MDA8 (Unit: μg/m³) in 2015 (a), 2016 (b), 2017 (c) and 2018 (d), relative to the mean during 2015–2018. The black pluses indicate that the maximum MDA8 was larger than 265 μg/m³.

**Figure 9: Are the variations in reference to an average for all available sites in the NC and the YRD regions? If so, please add this level of detail to the Figure 9 caption.**

*Reply:*

According to the reviewer's advice, this Figure caption was improved.

*Revision:*

**Figure 9.** Variations in the MDA8 (Unit: μg/m³) of NC (black) and the YRD (blue) in 2016 (a) and 2018 (b). **The MDA8 was calculated as an average for all available sites in the NC and the YRD regions.**

**Response to Reviewer #2**

The quality of the manuscript has been improved. The revised manuscript is much easier to follow, and the figures are carefully labeled and explained. Overall the authors have well addressed my concerns, but I'm confused with the authors' reply to my first comment regarding the impact of anthropogenic emissions.

**In the first point, the authors clarify they are more interested in the fluctuation of MDA8 ozone, but the inter-annual variations in emissions should also have impacts on the anomalies of MDA8 ozone.**

**In the second point, the authors argue that the emissions are stable at daily scale and it's hard to acquire daily emission data, but I don't think this argument justifies why the impacts of emissions on the inter-annual variability of ozone (i.e. Figure 7) are not accounted. I'd suggest the authors provide some discussions on how the recent decreasing trends of NOx emissions could impact on the inter-annual variability of ozone.**

*Reply:*

(1) As regards the mechanisms, our study mainly focus on the impacts of meteorological conditions on daily time-scale.

(2) The discussion of the impact of decreasing trends of NOx was really absent, which was strengthened in the revised manuscript and can be found in the following revisions.

(3) **According to the other reviewer's comment, we decided to delete the Figure 10 and 11, i.e., the iner-annual variability of ozone was not discussed in this revised version.** We now focus on the dominant patterns and their varying features in different years. Some new works will be supplemented and we will revisit the Figure 10 and Figure 11 in a later manuscript.

The contents related to Figure 7–9 were rewritten and redistributed in the revised version. The texts associated Figure 10 &11 were deleted. Detailed revisions can be found in the revised manuscript and also the mark-up manuscript.

*Revision in the Introduction:*

[revised manuscript text omitted]

---

## Author Response (AR3)

**Response to Editor**

*Comments:*

Please clarify the text on lines 84-90.

*Non-public comments to the Author:*

Overall the reviewer is happy with your responses and revisions, but would like just one more clarification:

"Overall, I'm satisfied with the minor corrections to my comments. The only one I still don't like is the way they explain the reanalysis timesteps. I still find this confusing.

I had to physically go to the ecmwf page and try to download the data to understand what they were talking about.....which means it is not explained well in the text....but maybe that's a level of detail that is not necessary. My issue is with the text on line 84 to 90 in the latest version. Some of the variables they use are available at the synoptic times, every 6 hours, and this they refer to as the '6-hourly reanalysis' data since these are analyzed fields. The '3-hourly data (precipitation and downward solar radiation)' is not referred to as reanalysis data since it is not constrained by observations, but forecasts from the 00Z and 12Z analysis times.

This 3-hourly data has

TIME = 0000/1200,

STEP = 3/6/9/12,

Choosing 21 to 09 UTC means they are using values from different forecasts (the +9 and +12 forecast from the 12Z, and then +3, +6, +9 forecasts from the 00Z) but since they are averaging over long periods, this probably doesn't matter.

So since the forecasts are available at the synoptic times (0, 6, 12, 18 UTC) I don't understand why they say "Due to the different representative period of each element in ERA-Interim data" and then choose different periods of time for "daytime"."

*Reply:*

We must apologize for our poor expressions to answer the abovementioned questions, although we tried our best to reply all the time. In this revised version, we

do more calculations to unify the daytime of different variables. That is, the daytime is from 05 to 17 (Beijing Time; 21–09 UTC) for all of the employed variables after revisions. Detailed explanations were as follows:

(1) As regards the ***Instantaneous Data*** (e.g., Z, wind, relative humidity, vertical velocity, air temperature and cloud cover), we calculated mean of values at 00:00 and at 06:00, which represent mean value from 21:00 to 09:00 (Fig R1 (b)), as daytime-average. So data daytime is 21 to 09 UTC (Beijing time 5:00 to 17:00). It is **the only daytime for summertime Beijing**.

[Figure]

Figure R1. ERA-Interim data time steps. The red bracket under X-axis represent and time scale we used. The red solid arc is steps we used. The red dashed arc mean that 21:00-00:00 is difference between 12:00-00:00 and 12:00-21:00.

(2) However, for ***accumulated precipitation and downward solar radiation***, we make more calculations to unify the daytime to 5:00 to 17:00 (Beijing time). The **sum of two time scale**, from 21:00 to 00:00 and from 00:00 to 09:00, used as daytime data now (Fig R1 (a)). Thereinto, 21:00-00:00 is ***calculated as the difference*** between '+12' from 12Z and '+9' from 12 Z. 00:00 to 09:00 is '+9' from 00Z.

After revisions, the difference between original Figures in last version and this revised version is negligible (Fig R2; Fig R3). **The comparisons in Figure R2 and R3 proved that the abovementioned changes do not affect our conclusion, but the daytime was unified and the confusions were avoided.**

[Figure]

Figure R2. The daytime atmospheric circulations (i.e., PAT1P minus PAT1N). (a) Precipitation accumulated from **00 UTC to 12 UTC**; (b) Precipitation accumulated from **21 UTC to 09 UTC**; (c) downward solar radiation accumulated from 00 UTC to 12 UTC; (d) downward solar radiation accumulated from 21 UTC to 09 UTC. The green boxes show the NCH region.

[Figure]

Figure R3. The daytime atmospheric circulations (i.e., PAT2P minus PAT2N). (a) Precipitation accumulated from 00 UTC to 12 UTC; (b) Precipitation accumulated from 21 UTC to 09 UTC; (c) downward solar radiation accumulated from 00 UTC to 12 UTC; (d) downward solar radiation accumulated from 21 UTC to 09 UTC. The green boxes show the NC and YRD regions.

*Revision in Data Description:*

[revised manuscript text omitted]